# Conformational flexibility in neutralization of SARS-CoV-2 by naturally elicited anti-SARS-CoV-2 antibodies

Ruofan Li[1,8], Michael Mor [2,8], Bingting Ma[1,8], Alex E. Clark [3], Joel Alter[4], Michal Werbner[5], Jamie Casey Lee [6], Sandra L. Leibel[6,7], Aaron F. Carlin [3], Moshe Dessau [4], Meital Gal-Tanamy[5], Ben A. Croker [6✉], Ye Xiang [1✉] & Natalia T. Freund [2✉]

As new variants of SARS-CoV-2 continue to emerge, it is important to assess the cross-neutralizing capabilities of antibodies naturally elicited during wild type SARS-CoV-2 infection. In the present study, we evaluate the activity of nine anti-SARS-CoV-2 monoclonal antibodies (mAbs), previously isolated from convalescent donors infected with the Wuhan-Hu-1 strain, against the SARS-CoV-2 variants of concern (VOC) Alpha, Beta, Gamma, Delta and Omicron. By testing an array of mutated spike receptor binding domain (RBD) proteins, cell-expressed spike proteins from VOCs, and neutralization of SARS-CoV-2 VOCs as pseudoviruses, or as the authentic viruses in culture, we show that mAbs directed against the ACE2 binding site (ACE2bs) are more sensitive to viral evolution compared to anti-RBD non-ACE2bs mAbs, two of which retain their potency against all VOCs tested. At the second part of our study, we reveal the neutralization mechanisms at high molecular resolution of two anti-SARS-CoV-2 neutralizing mAbs by structural characterization. We solve the structures of the Delta-neutralizing ACE2bs mAb TAU-2303 with the SARS-CoV-2 spike trimer and RBD at 4.5 Å and 2.42 Å resolutions, respectively, revealing a similar mode of binding to that between the RBD and ACE2. Furthermore, we provide five additional structures (at resolutions of 4.7 Å, 7.3 Å, 6.4 Å, 3.3 Å, and 6.1 Å) of a second antibody, TAU-2212, complexed with the SARS-CoV-2 spike trimer. TAU-2212 binds an exclusively quaternary epitope, and exhibits a unique, flexible mode of neutralization that involves transitioning between five different conformations, with both arms of the antibody recruited for cross linking intra- and inter-spike RBD subunits. Our study provides additional mechanistic understanding about how antibodies neutralize SARS-CoV-2 and its emerging variants and provides insights on the likelihood of reinfections.

[1] Beijing Advanced Innovation Center for Structural Biology, Beijing Frontier Research Center for Biological Structure, Center for Infectious Disease Research, School of Medicine, Tsinghua University, Beijing, China. [2] Department for Microbiology and Clinical Immunology, Faculty of Medicine, Tel Aviv University, Tel Aviv, Israel. [3] Department of Medicine, University of California San Diego, La Jolla, CA, USA. [4] The Laboratory of Structural Biology of Infectious Diseases, Azrieli Faculty of Medicine, Bar Ilan University, Tsafed, Israel. [5] Molecular Virology Lab, Azrieli Faculty of Medicine, Bar Ilan University, Tsafed, Israel. [6] Department of Pediatrics, School of Medicine, UC San Diego, La Jolla, CA, USA. [7] Sanford Burnham Prebys Medical Discovery Institute, La Jolla, CA, USA. [8] These authors contributed equally: Ruofan Li, Michael Mor, Bingting Ma. ✉email: bcroker@health.ucsd.edu; yxiang@mail.tsinghua.edu.cn; nfreund@tauex.tau.ac.il

Within 2 years of the emergence of SARS Coronavirus 2 (SARS-CoV-2) in Wuhan province, the original virus strain has been completely replaced by more transmissible variants, with Omicron emerging as the latest variant of concern (VOC). In view of the unexpectedly fast rates of viral evolution, it is important to estimate the degree to which neutralizing antibodies elicited naturally following infection with the original wild type strain (Wuhan-Hu-1), are cross reactive with circulating, present and future, VOCs. This is particularly relevant considering recent reports that vaccination provides considerably less protection against SARS-CoV-2 variants than against the original strain[1–3].

The SARS-CoV-2 antibody response has been profiled at the sequence, structural, and mechanistic level by cloning and characterizing monoclonal antibodies (mAbs) from Wuhan-Hu-1-infected convalescent donors[4–9]. However, with the emergence of variants, many of these mAbs, some of which have been approved for treatment of COVID-19 patients[10–12], have become ineffective[13–15], while others retain activity[16]. This indicates that some antibodies elicited by infection are more variation-sensitive than others, and that antibody breadth of specificity, and not only potency, should be considered. A finer resolution investigation of the mechanistic and functional basis of SARS-CoV-2 antibody neutralization is therefore needed to predict the effect viral modifications will have on antibody activity, and to estimate the degree of protection from SARS-CoV-2 reinfection and breakthrough infection. Moreover, studying the molecular recognition of naturally elicited neutralizing antibodies in the context of heterologous viruses can reveal sites with a lower tendency to variation.

We have previously identified a panel of neutralizing SARS-CoV-2 antibodies derived from two COVID-19 survivors who were infected in Israel in March 2020, likely with the Wuhan-Hu-1 strain[9]. Seven of these mAbs (TAU-1109, -1145, -2189, -2212, -2230, -2303, and -2310) exhibited potent SARS-CoV-2 neutralizing activity, while the activity of another two (TAU-1115 and TAU-2220) was less potent. All the mAbs, except one, bind soluble receptor binding domain (RBD) and spike with high affinity. The exception, TAU-2212, which is one of the most potent neutralizing mAbs, binds an unknown conformational surface, and binding can only be detected when the spike protein is expressed on viral particles or cells. While exhibiting neutralizing activity against the Wuhan-Hu-1 strain, the cross-reactivity of these mAbs with SARS-CoV-2 VOCs is unknown.

The present study was designed to investigate the breadth of specificity and structural basis of the neutralizing activity of our previously isolated mAbs, in the context of the emerging variants Alpha (B.1.1.7)[17], Beta (B.1.351)[18], Gamma (P.1 or B.1.1.28.1)[19], Delta (B.1.617.2)[20,21] and Omicron (B.1.1.529)[22]. The results indicate that the most potent mAbs in our panel are predominantly directed against the ACE2 binding site (ACE2bs) supersite and are also the ones most sensitive to viral diversification. To understand the basis of neutralization and escape at the atomic level, we used cryo electron microscopy (cryoEM) and X-ray crystallography to determine structures of the two mAbs: TAU-2303 and TAU-2212. The results indicate that the interaction of TAU-2303 Fab and SARS-CoV-2 RBD resembles that of the ACE2 receptor with RBD in the "up" RBD conformation. In contrast, mAb TAU-2212 exhibits an unusual recognition flexibility type of binding involving five different possible conformations, with 1–3 Fabs binding to one spike trimer, or cross-linking adjacent trimers by forming both intra- and inter-spike contacts, and favoring RBD in the "down" position. Our study provides important mechanistic and structural insights about neutralization of SARS-CoV-2 VOCs by natural antibodies, together with molecular modeling predictions of mAb interactions with the Omicron variant.

## Results

**Antibodies binding at the ACE2bs are more sensitive to viral mutations**. The results of our previous study indicated that mAbs TAU-1145, -2189, -2230, and -2303 compete with ACE2, and are therefore defined as ACE2bs mAbs, while mAbs TAU-1109, -1115, -2220, -2310 do not compete with ACE2, and are therefore defined as non-ACE2bs[9]. The last neutralizing mAb, TAU-2212, does not bind soluble SARS-CoV-2 antigens (RBD or spike) by ELISA and recognizes an unknown epitope[9]. To examine the recognition of the RBDs from SARS-CoV-2 VOCs by our previously isolated mAbs, we generated soluble RBD proteins containing the mutations identified in the Alpha, Beta, Gamma, Delta and Omicron strains (Supplementary Table 1) and tested the binding by the eight mAbs that originally exhibited strong binding to the wild type strain RBD[9]. With the exception of Alpha, we observed a reduction in binding efficiency of the ACE2bs mAbs to all the VOCs (Fig. 1a and Supplementary Fig. 1). This was most significant for Beta, Delta and Omicron RBDs but was also present, albeit to a lesser extent, for the Gamma variant. Amongst the ACE2bs mAbs, only TAU-2303 maintained its original activity against the Delta strain. TAU-2212 does not bind soluble RBD and could not be evaluated using this assay[9]. To investigate the individual contribution of each mutation, we generated RBDs harboring single or double amino acid substitutions corresponding to variants Beta, Gamma and Delta. Of the eight single-mutated RBDs tested, both the L452R (present in the Delta VOC[23]) and E484K (present in both the Beta and Gamma VOCs[24]) substitutions had a major impact on antibody binding (Fig. 1b, d and Supplementary Fig. 1). Other single substitutions, however, had no effect on mAb binding. Furthermore, RBDs containing single mutations N439K[25], Y453F[26,27] and A475V[28], which have been reported in some circulating SARS-CoV-2 strains, were bound by all mAbs as strongly as the original wild type RBD. Interestingly, the binding of mAb TAU-2303 to the double mutant K417N/N501Y was reduced although each of the two single mutants separately had no effect (Fig. 1c, d and Supplementary Fig. 1). Overall, we conclude that ACE2bs mAbs are more sensitive to mutations in the RBDs than non-ACE2bs mAbs.

Mutations outside the RBD may also affect the activity of RBD-binding mAbs by altering the conformational organization of the trimer[29]. Therefore, we next expressed the full-length spike protein of wild type, Alpha, Beta, and Delta variants (Supplementary Table 1) on Expi293F cells, and assayed the ability of each mAb to inhibit binding of soluble human ACE2 (hACE2) by flow cytometry. The Gamma full-length spike protein was not produced since the RBD of this variant exhibited similar activity to the Beta RBD. As expected, most of the ACE2bs mAbs effectively inhibited the ACE2:spike$_{wild type}$ and ACE2:spike$_{Alpha}$ interactions, but not those between the ACE2:spike$_{Beta}$ and ACE2:spike$_{Delta}$. The mAb TAU-2212, which can be tested in this assay, demonstrated 25–40% ACE2:spike inhibition when tested against the wild type, Alpha and Delta strains, but had no activity against Beta (Fig. 2a, b and Supplementary Fig. 2). In fact, none of the ACE2bs mAbs retained their original potency against the Beta variant, and, in accordance with the ELISA data, only mAb TAU-2303 retained its full activity against the Delta variant. As expected, no effect was observed for mAbs TAU-1109, -2310, -1115 and -2220, as the neutralization does not act through receptor blocking.

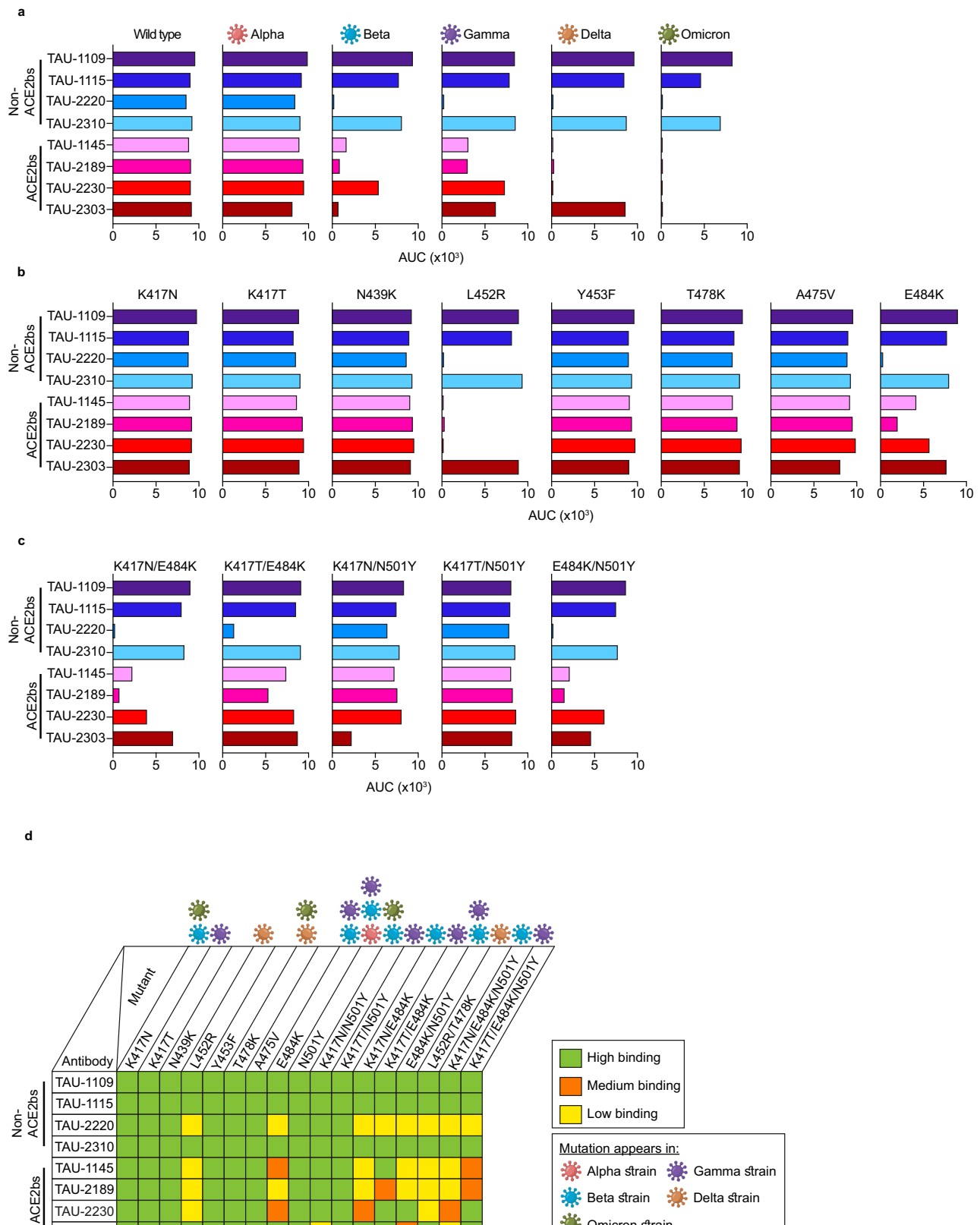

**Fig. 1 Antibody binding to the RBD of wild type and SARS-CoV-2 VOCs by ELISA.** ACE2bs or non-ACE2bs antibodies are indicated to the left of each panel, **a–d**. For panels **a–c**, each graph represents antibody binding to wild type or VOC (**a**), single (**b**) or double (**c**) mutant RBD. AUC was calculated by GraphPad Prism. The raw $OD_{650}$ values, as well as isotype control, are presented in Supplementary Fig. 1. Each experiment was repeated at least three times ($n \geq 3$). **d** Summary of the antibody binding affinity to each RBD generated in this study. Green color indicates binding affinity of >75%, orange of 25–75%, and yellow of <25% as compared to wild type RBD. The VOCs harboring each mutation are indicated.

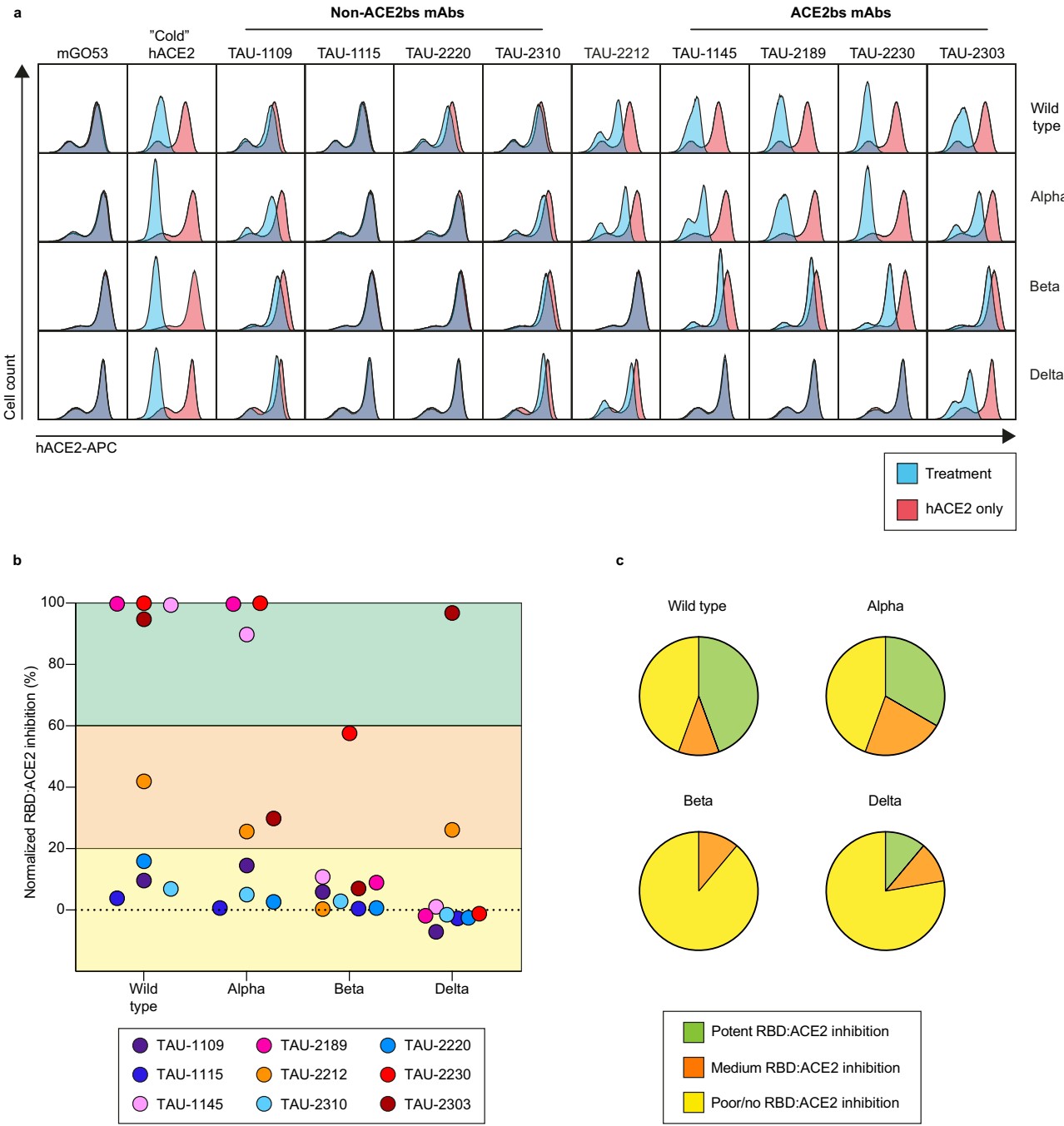

**Fig. 2 Antibody inhibition of soluble human ACE2 binding to cell-expressed spike. a** Flow cytometry plots demonstrating the effectiveness of each mAb when interfering with spike:ACE2 binding (see Supplementary Fig. 2 for more details). Expi293F cells were transfected to express the wild type, Alpha, Beta, or Delta spike, and were incubated with each antibody, before being stained with human ACE2 (hACE2) conjugated to APC. Unlabeled hACE2 ("cold" hACE2) was used as a positive control, and mGO53[66] antibody as an isotype control. Within each plot, the blue histogram indicates treated cells, while the red indicates untreated. Each experiment was repeated at least three times ($n \geq 3$). **b** Normalized percent of spike:ACE2 inhibition, calculated by measuring the percent of hACE2-APC positive cells in the presence of each mAb, dividing it by the percent of hACE2-APC positive cells (hACE2 only), and normalizing to 100%. **c** Pie charts indicating the frequency of spike:ACE2 inhibiting mAbs for wild type and VOCs.

**SARS-CoV-2 VOCs are resistant to most ACE2bs mAbs**. We next evaluated the ability of the mAbs to prevent infection. We employed a pseudo-viral neutralization assay and infection of Vero-TMPRSS2 cells with authentic SARS-CoV-2 to test the activity of the nine mAbs against SARS-CoV-2 wild type and VOCs (Fig. 3). In accordance with the ELISA and flow cytometry results, the Alpha VOC behaved similarly to the wild type strain, while the Beta and Omicron variants were the most resistant, followed by Gamma and Delta. In agreement with the ELISA and

flow cytometry results, mAb TAU-2303 was the only ACE2bs mAb that was able to neutralize the Delta VOC, with improved potency compared to the wild type strain (Fig. 3a). The non-ACE2bs mAbs TAU-1109 and -2310 retained their efficacy against all tested VOCs, with TAU-2310 exhibiting improved activity against the Delta variant compared to the wild type (Fig. 3a, b). These results indicate a crucial support role for non-ACE2bs mAbs in the presence of viral mutations that prevent neutralization by ACE2bs mAbs. Additionally, the improved

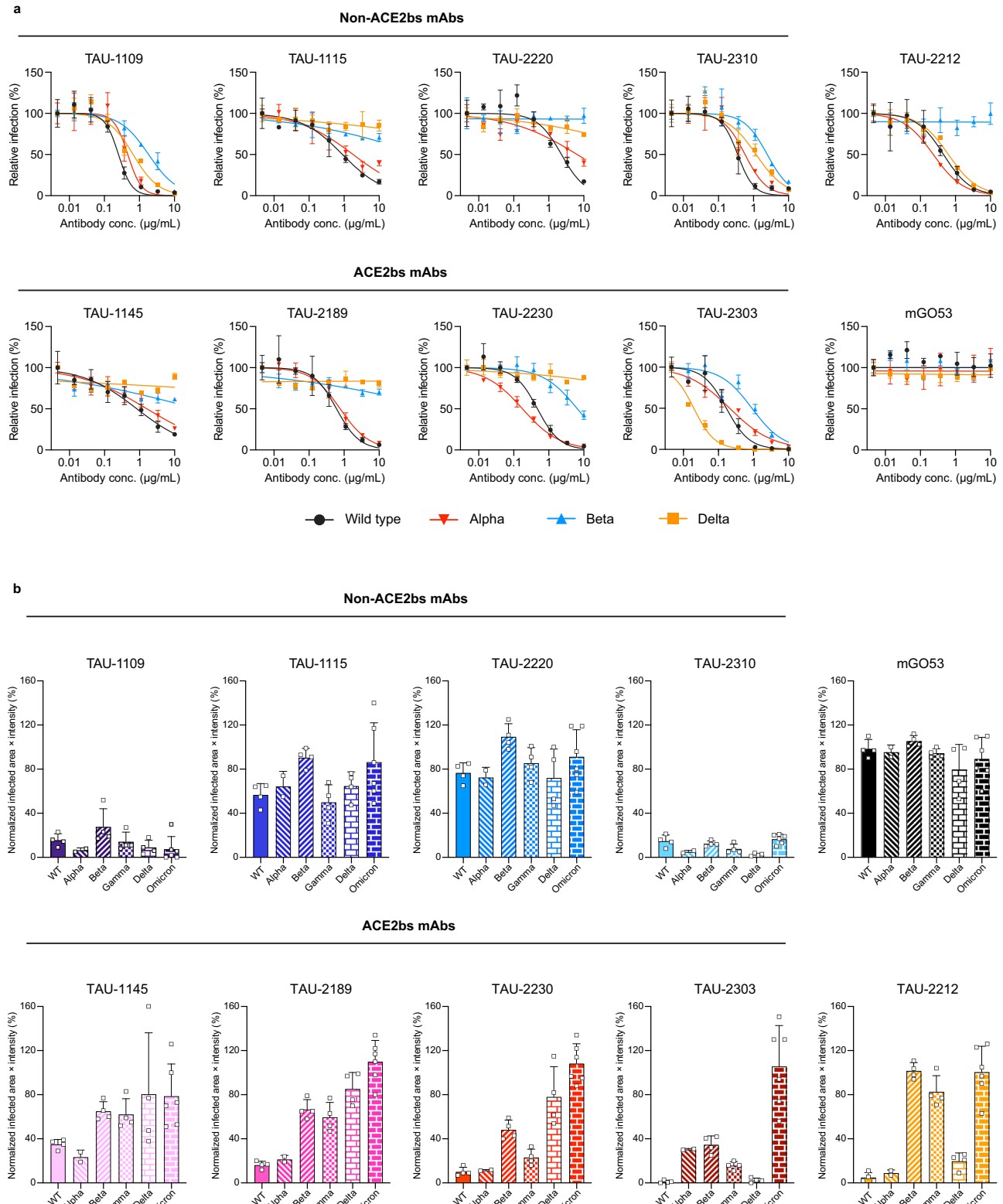

**Fig. 3 Antibody neutralization of wild type and VOCs. a** Antibody neutralization curves of wild type, Alpha, Beta, and Delta VOC pseudo-typed GFP reporter viral particles. For every mAb, each curve represents inhibition of one VOC as indicated. Antibodies were pre-incubated with viral particles at 8 consecutive 3-fold dilutions starting at 10 μg/mL of antibody. Fluorescence of infected cells was read 72 h post infection. Inhibition percentage was calculated by normalizing to untreated cells. Each experiment was done in triplicate ($n = 3$). mGO53 was used as an isotype control. Error bars indicate SD. **b** Infection of Vero-TMPRSS2 cells by wild type ($n = 4$), Alpha ($n = 2$), Beta ($n = 4$), Gamma ($n = 4$), Delta and Omicron ($n = 6$) with authentic SARS-CoV-2 and overlayed with carboxymethylcellulose. Values are expressed as the infected surface area of cells and normalized to infected cells without mAb. Viral particles were pre-incubated with 100 μg/mL of antibody for 1 h before addition to the cells. Infected cells were identified at 24 h post infection using SARS-CoV-2 nucleocapsid antibody after cell fixation and permeabilization. Infected cells were quantified using Incucyte S3. mGO53 serves as an isotype control. Error bars indicate SD.

**Structural basis of neutralization of the ACE2bs mAb TAU-2303**. Viral evolution appears to be primarily focused on the ACE2bs supersite, consistent with the overall superoir potency of receptor blocking mAbs to inhibit SARS-CoV-2. We decided to focus on mAb TAU-2303, which is the only ACE2bs mAb that is active against the Delta VOC, and mAb TAU-2212 that blocks receptor binding through recognition of a conformational surface, and further investigate the structural basis of neutralization for these two mAbs. We first determined the cryoEM structure of the fragment of antigen-binding (Fab) of TAU-2303 (Fab2303) in complex with the ecto SARS-CoV-2 spike trimer, at a resolution of 4.5 Å (Fig. 4a). The cryoEM structure revealed that one Fab2303 molecule binds one spike trimer on the protruding "up" facing RBD. Therefore, mAb TAU-2303 can be categorized as a CoV21[30]-type mAb, that belongs to Class 1 RBD binding mAbs, and binds only one RBD subunit amongst the three available subunits of the trimer (Fig. 4a). We also crystallized Fab2303 in complex with SARS-CoV-2 RBD and analyzed the structure at a resolution of 2.42 Å (Fig. 4b Table 1). The results from both the crystal and cryoEM structures indicated that Fab2303-RBD complex has a large buried surface of 1185 Å$^2$, with the majority of the contact surface (64%) derived from the heavy chain of Fab2303, and only 36% from the light chain. A total of 29 RBD residues in the crystal structure have direct close contacts with Fab2303 (Supplementary Table 2). Consistent with the contact surface analysis, 19 of these residues are from the heavy chain (HC), while 10 are from the light chain (LC) of TAU-2303. The contact residues mediate the formation of 23 hydrogen bonds (Supplementary Table 2). These include five hydrogen bonds RBD-D420$^{OD2}$ - Fab2303-HC-S56$^{OG}$, RBD-Y473$^{OH}$ - Fab2303-HC-S31$^O$, RBD-T415$^{OG1}$ - Fab2303-HC-Y58$^{OH}$, RBD-L455$^O$ - Fab2303-HC-Y33$^{OH}$, and RBD-Q493$^{NE2}$ - Fab2303-HC-Y102$^{OH}$ with distances of below 2.7 Å (Supplementary Table 2). In accordance with the biochemistry and in vitro cell assays, 14 (K417, T453, L455, A475, F486, N487, Y489, Q493, G496, Q498, T500, N501, G502, and Y505) of the 29 contacts between Fab2303 and RBD, are also involved in binding the ACE2 receptor, thus confirming that TAU-2303 neutralizes the virus by blocking receptor binding (Figs. 1–3, 4b and Supplementary Figs. 1,2). The angle of approach through which Fab2303 binds RBD is 25° relative to that of ACE2 (Fig. 4b). Further comparisons revealed that the binding mode of mAb TAU-2303 is similar to that of other Class 1 neutralizing antibodies that target RBD (Supplementary Fig. 3a, b)[30–32].

The ELISA results indicated a reduction in TAU-2303 binding to RBD with the double mutations K417N/N501Y, but not to K417T/N501Y, or any of the single mutations K417N, K417T, or N501Y (Fig. 1 and Supplementary Fig. 1). This agrees with the crystal structure findings that residues K417 and N501 are both in direct contact with TAU-2303. Position 417 is particularly critical for TAU-2303 recognition, as Fab2303-HC-Y52$^{OH}$ forms hydrogen bonds both with the main chain atom N and the side chain atom NH of RBD-K417 (Fig. 4c, d and Supplementary Table 2). Modeling revealed that when lysine in position 417 is replaced with threonine, both the main chain and side chain hydrogen bonds could be retained (Fig. 4d). Moreover, it becomes possible to establish a new hydrogen bond between the OH atom of the threonine and the OH atom Fab2303-HC-Y33, thus decreasing the total binding energy calculated with PISA[33] from -9.7 kJ/mol to -9.9 kJ/mol. In contrast, replacing lysine in position 417 by asparagine, increases the length of the hydrogen bonds between

the side chain polar atoms ND2H or OD1 with Fab2303-HC-Y33 and Fab2303-HC-Y52, which decreases the favorability of the interaction, as indicated by the increase of the calculated binding energy from -9.7 kJ/mol to -9.5 kJ/mol (Fig. 4d). The asparagine in position 501 has a relatively large number of contacts with Fab2303. The N501 side chain forms two hydrogen bonds with main chain and side chain atoms of Fab2303-LC-S30 (Fig. 4c). While these hydrogen bonds would be completely disrupted by replacing asparagine with tyrosine, our modeling data suggest that new Van der Waals interactions could be formed with tyrosine in this position (Fig. 4d). Since these proposed Van der Waals interactions would not be able to fully compensate for the lost hydrogen bonds, mutant N501Y is predicted to have a slightly lower binding affinity to Fab2303, as is confirmed by affinity measurments by Surface Plasmon Resonance (SPR) (Supplementary Fig. 3c, d). Interestingly, although glutamate in position 484 was shown to be central to viral escape from neutralizing antibodies[34–36], our structural analysis indicates that E484 does not make direct interactions with Fab2303. This point was resolved by further examination of the RBD and Fab2303 interface, which revealed that E484 interacts with Fab2303-HC-Y102 through a water molecule (Fig. 4d). The observation that solvent-mediated interactions are usually weaker than direct contacts, is consistent with the results of our functional studies that, in contrast to the other ACE2bs mAbs, the single E484K mutation had no significant impact on TAU-2303 binding (Fig. 1 and Supplementary Fig. 1). The mutations T487K and L452R of RBD$_{Delta}$ are located outside the binding epitope of Fab2303, which, thus explains how TAU-2303 can still bind with high affinity and neutralize the Delta VOC (Fig. 4e and Supplementary Fig. 3c, d). The structure and modeling data explains why TAU-2303 can bind equally well to an RBD molecule with a single mutation, such as K417N, E484K, and N501Y, but not to an RBD mutant with all three mutations, with the greater cumulative loss of binding energy. Our structural data combining the biochemistry and neutralization results, suggest that the combination of RBD mutations may play a key role in conferring SARS-CoV-2 resistance. Significantly, the Omicron variant, has mutations in seven residues within the binding epitope of Fab2303, including four key contacting residues K417, Q493, Q498, and Y505, providing the structural explanation to the lack of neutralization of Omicron by TAU-2303 (Figs. 3 and 4e).

**mAb TAU-2212 blocks ACE2 binding through conformational dynamics**. TAU-2212 is one of the most potent mAbs in our panel, and while not being able to bind soluble RBD by ELISA, it inhibits ACE2 binding as measured by flow cytometry, suggesting that receptor binding is blocked through a different mechanism to that employed by TAU-2303. To investigate the neutralization mechanism of TAU-2212, we prepared Fab2212 by cleaving mAb TAU-2212 with papain. The purified Fab2212 was crystallized and the crystal was diffracted to 2.7 Å. Data analysis revealed that the crystal belongs to the space group P6$_5$ with cell dimensions $a = 75.98$ Å, $b = 75.98$ Å, and c = 348.14 Å (Table 1), and with two molecules in the asymmetric unit. The structure was determined by molecular replacement and was refined to a final R/Rfree of 0.202/0.246. The final model of Fab2212 contains 419 residues, while residues 141–151 and 197–204 of the heavy chain are not visible in the map (Fig. 5a). No significant differences were observed between the two molecules in the asymmetric unit (a calculated R.M.S.D. of 0.77 Å between the aligned C$_{alpha}$ atoms, Supplementary Fig. 4). Fab2212 is not able to form a stable complex with the prefusion stabilized SARS-CoV-2 spike ecto-domain (Supplementary Fig. 5a), but interacts with spike trimer on the cell surface[9]. With the supposition that TAU-2212 binds a

activity of TAU-2310 and TAU-2303 mAbs demonstrates how genetic variation in a SARS-CoV-2 VOC can increase neutralization for some classes of mAbs.

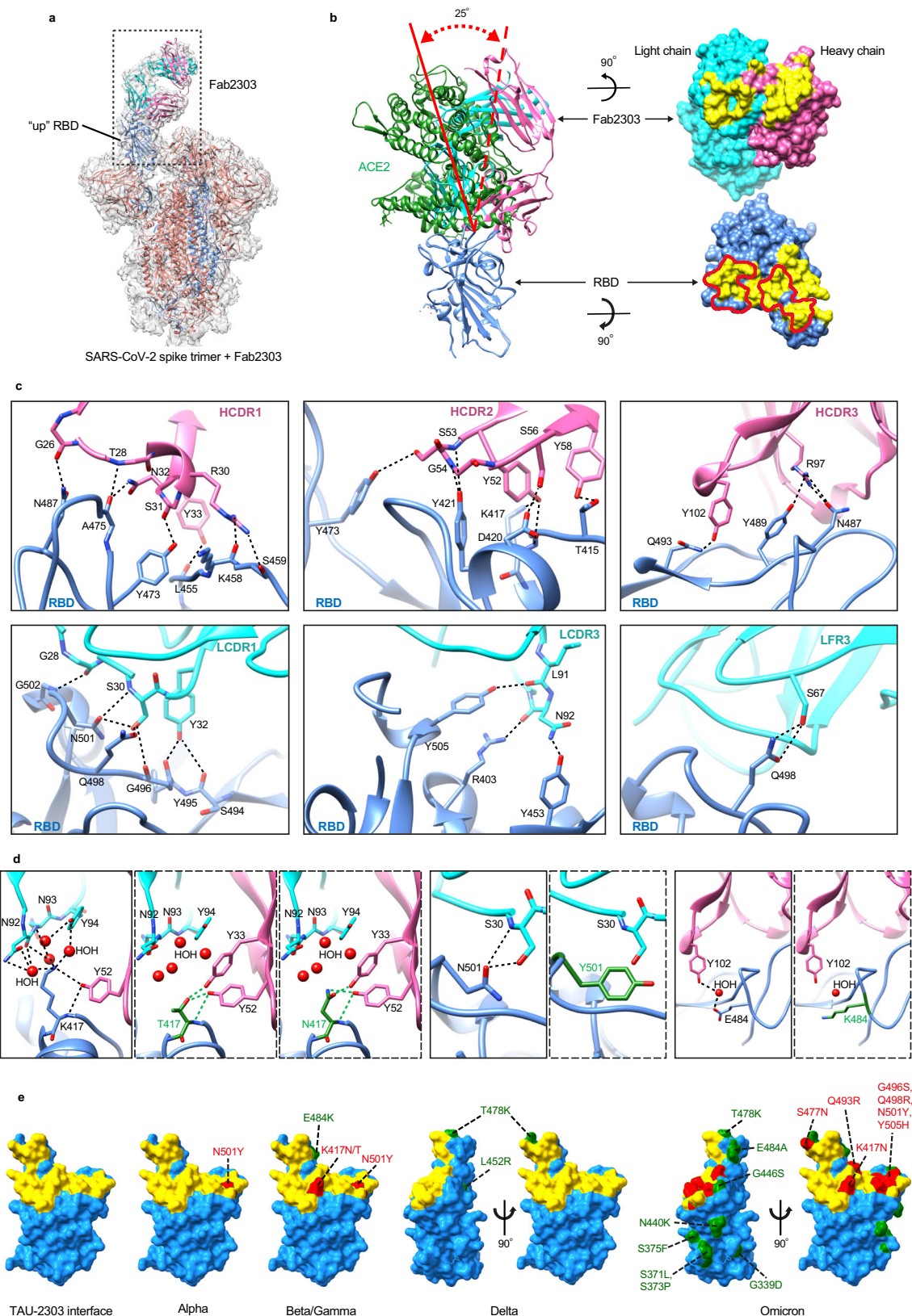

specific conformation of the prefusion spike trimer that is not stable when coated on ELISA plates, we used TAU-2212-protein A coated beads to pull down spike trimers, which were then subjected to cryoEM structural analysis. CryoEM 3-dimensional (3D) classification of the particles revealed that TAU-2212 binds the spike trimer in five distinct conformations (1–5), with 20339,

15868, 14193, 72056, and 39788 particles, respectively (Supplementary Fig. 6 and Table 2). In all five conformations, only the Fab portion of mAb TAU-2212 is visible, while the highly flexible Fc region is not seen in the reconstructed maps. Conformations 1, 3 and 4 are composed of two, three and three bound Fabs, respectively, with all the RBDs in the "down" position. In

**Fig. 4 Structural analysis of Fab2303 in complex with the SARS-CoV-2 spike trimer and RBD. a** Ribbon diagrams showing the cryoEM structure of the SARS-CoV-2 spike trimer in complex with one TAU-2303 Fab (Fab2303). The "up" RBD protomer of the spike is colored blue. The other two protomers are colored salmon. The heavy chain of Fab2303 is colored pink and the light chain is colored cyan. **b** Left: ribbon diagrams showing the Fab2303-RBD crystal structure superimposed onto the ACE2-RBD crystal structure (PDB: 6M0J). The RBD and Fab2303 are colored as in **a**. ACE2 is colored green. The solid and dashed lines in red indicate the long axes of ACE2 and Fab2303, respectively. Right: the paratope and epitope of Fab2303 shown as rendered surface representations. The paratope on Fab2303 and epitope on RBD are colored yellow. Red lines indicate the footprint of ACE2. **c** Detailed interactions between Fab2303 and SARS-CoV-2 RBD. HCDR and LCDR stand for complementarity-determining region (CDR) of the heavy and the light chain, respectively. LFR stands for framework region of the light chain. **d** Structural comparisons of RBD with RBD mutants K417N, K417T, N501Y, and E484K. Structures of the mutants were modeled in COOT[53] by using the single mutate function, in which only the side chains of the mutated residues were changed. Most possible side chain conformations of the mutated residues were generated and selected from the rotamer library of COOT and according to the binding energy calculated with PISA. The heavy chain of Fab2303 is colored pink and the light chain is colored cyan. The RBD and Fab2303 are colored as in **a**, with the mutated RBD residues in green. **e** Surface mapping of key mutations in different variants and the positions of the mutated sites relative to the binding epitopes recognized by TAU-2303. The binding epitopes of TAU-2303 are colored yellow. The mutation sites within or outside the binding epitope of TAU-2303 are colored red and green, respectively.

**Table 1 X-ray Data collection and refinement statistics for RBD–Fab2303 complex and Fab2212.**

|  | RBD–Fab2303 | Fab2212 |
|---|---|---|
| **Data collection** | | |
| Space group | C 2 2 2$_1$ | P 6$_5$ |
| **Cell dimensions** | | |
| $a$, $b$, $c$ (Å) | 85.13, 149.98, 144.79 | 75.98, 75.98, 348.14 |
| $\alpha$, $\beta$, $\gamma$ (°) | 90, 90, 90 | 90, 90, 120 |
| Resolution (Å) | 20.09–2.42 | 29.75–2.71 |
|  | (2.51–2.42) * | (2.80–2.71) * |
| $R_{merge}$ | 0.19 (0.751) | 0.145 (0.924) |
| $I/\sigma I$ | 12 (2) | 25 (2.9) |
| Completeness (%) | 99.44 (95.98) | 99.68(97.60) |
| Redundancy | 12.1 (10.2) | 8.5 (7.1) |
| **Refinement** | | |
| Resolution (Å) | 20.09–2.42 | 29.75–2.71 |
| No. reflections | 35382 (3389) | 30759 (3006) |
| $R_{work}/R_{free}$ | 0.1821/0.2201 | 0.2015/0.2462 |
| No. atoms | 5171 | 6626 |
| Protein | 4821 | 6293 |
| Ligand/ion | 14 | – |
| Water | 336 | 333 |
| $B$-factors(Å$^2$) | 47.08 | 93.79 |
| Protein | 46.99 | 95.59 |
| Ligand/ion | 72.98 | – |
| Water | 47.24 | 59.67 |
| **R.m.s. deviations** | | |
| Bond lengths (Å) | 0.004 | 0.008 |
| Bond angles (°) | 0.72 | 0.97 |

*Values in parentheses are for highest-resolution shell.

contrast, conformation 2 has one Fab per trimer, with one RBD in the "up" configuration, while the other RBDs that are bound by the Fab appear in the "down" configuration. Conformation 5 has two head-to-head spike proteins, which are crosslinked by three mAbs (Fig. 5b, c).

We determined the structures of all five conformations to resolutions of 4.7 Å, 7.3 Å, 6.4 Å, 3.3 Å and 9.4 Å (6.1 Å for the split single spike with three Fabs), respectively (Fig. 5b, c, Table 2 and Supplementary Figs. 6,7). For conformation 4, where the resolution permits ab initio model building and refinement (Supplementary Fig. 8), an atom model of the complexes was built with the crystal structure of Fab2212 and the cryoEM structure of the spike trimer (PDB: 6XEY) as references (Table 2). Since the resolution of the maps of conformations 1–3 and 5, is too low for ab initio model building, atom models of the complexes were built by fitting the crystal structure of Fab2212 and the cryoEM structure of the S trimer into cryoEM maps.

Conformation 4 of the TAU-2212:spike complex has all the RBDs in the "down" position with three TAU-2212 moieties binding near the junctions between the RBDs. Each Fab2212 binds and crosslinks two adjacent "down" RBDs (RBD1 and 2) with a buried surface of 823 Å$^2$ on RBD1 and 298 Å$^2$ on RBD2 (Fig. 5b). Comparing the structures of the TAU-2212-bound and free spikes revealed that TAU-2212 binding induces conformational changes in the RBDs, producing an anti-clockwise rotation of 4.1° towards the symmetry axis, and resulting in a more compact RBD head structure after TAU-2212 binding (Supplementary Fig. 9a). The conformational changes in the RBDs bring residues at the interfaces closer and may increase the Van der Waals interactions between the RBDs. However, no additional contacts are observed between the "down" RBDs. In addition, the RBD loops 454–462 and 468–489 that are disordered in free spikes are well ordered in the complex structure by forming three hydrogen bonds with the bound TAU-2212 (Supplementary Fig. 9b and Supplementary Table 3). These loops are also ordered and visible upon ACE2 binding. These data suggest that TAU-2212 acts by binding and stabilizing the SARS-CoV-2 spike trimer in a "down" orientation and prevents conformational change to "up" RBD which is required for spike:ACE2 interactions.

Most of the epitope recognized by mAb TAU-2212 is on one of the two RBDs (RBD1), and the interactions are predominantly through the heavy chain, especially the CDR3 loop of the heavy chain (HCDR3), which is embedded within the interface of two RBDs. Two CDR loops of the light chain interact directly with two residues on RBD1 (485–486), including the hydrogen bond formed between the N atom of F486$_{RBD1}$ and the OH group of Fab2212-HC-Y93 (Fig. 5d). Overall, the interactions between the HCDR loops and RBD1 and RBD2 involve 57 residues, and 13 pairs of hydrogen bonds (Fig. 5d and Supplementary Table 3). These interactions include two hydrogen bonds formed by the OE2 group of E484$_{RBD1}$, with the OH group of Fab2212-HC-Y33, and the ND2 group of Fab2212-HC-N52. The length of the hydrogen bond between the OE2 group of E484$_{RBD1}$ and the OH group of Fab2212-HC-Y33 is 2.2 Å, suggesting that this hydrogen bond could be the main interaction at the contact interface. The E484K mutation completely disrupts the hydrogen bonds between the antibody and the RBD (Fig. 5e and Supplementary Fig. 5b), providing a structural explanation for the complete resistance of the Beta and Gamma VOCs to TAU-2212. Similarly, both the E484A and Q493R mutations present in the Omicron variant, are considered likely to disrupt the key hydrogen bonds and are therefore expected to affect TAU-2212 binding and reduce neutralization capacity (Fig. 5f). Substitution S373P that is present in the new Omicron variant, also lies within the TAU-2212 interface. To test the effect of S373P mutation on TAU-2212

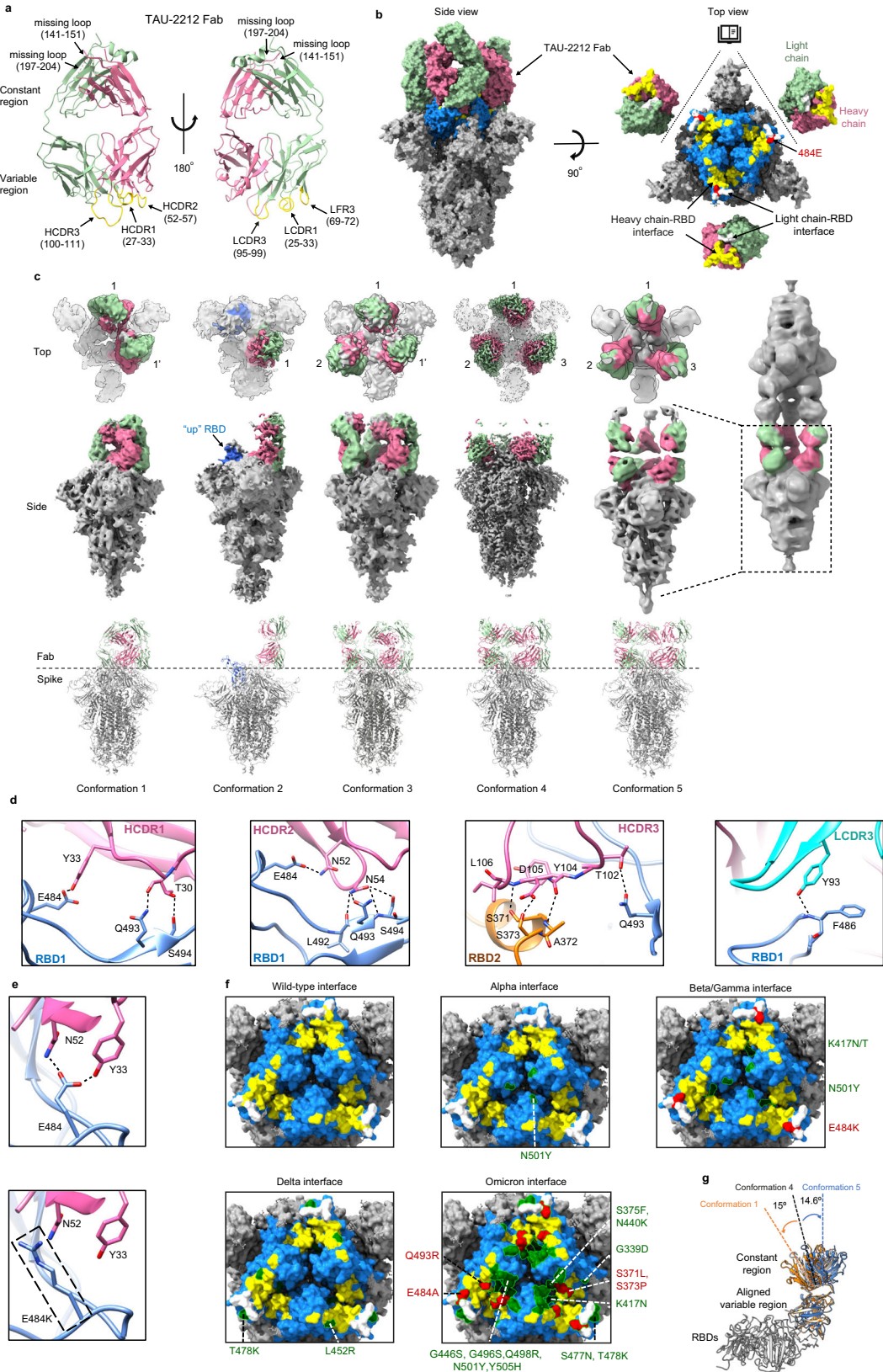

binding, we first introduced a S373G mutation to disrupt the hydrogen bond mediated by the hydroxyl group of serine 373. The results showed that the S373mediated hydrogen bond did not affect the binding of TAU-2212 (Supplementary Fig. 5b). Next, we performed a pull down experiment to test the interaction between mutant S373P and TAU-2212 (Supplementary Fig. 5c).

The results showed that TAU-2212 can still bind spike with the S373P mutation, which further supports that both the hydrogen bond and Van Der Waals force mediated by S373 side chain do not play a major role in TAU-2212 binding spike. Taking the effect of all the mutations together, it is therefore not surprising that TAU-2212 does not neutralize Omicron.

**Fig. 5 Structural analysis of TAU-2212 in complex with the SARS-CoV-2 spike trimer. a** Ribbon diagram showing the crystal structure of Fab2212. The heavy and light chains are colored pink and light green, respectively. The CDR loops are colored yellow. The missing residues 141–151 and 197–204 of the heavy chain are shown as pink dashed lines. **b** Surface diagrams showing the side (left) and the top open-up (right) views of the spike trimer with three bound Fabs of mAb TAU-2212. The RBDs of the spike trimer are colored blue and the Fab2212 heavy and light chains are colored as in **a**. The binding epitopes of the heavy and light chains around the junction of two RBDs are colored yellow and white, respectively. Residue E484 is indicated in red. **c** Surface (top) and ribbon (bottom) diagrams showing the five conformations of the TAU-2212-spike complex. The heavy and light chains of the bound mAbs are colored are colored as in **a**. The spike trimers are in gray. The number pairs X, X' indicate Fabs from the same mAb. **d** Detailed interactions between TAU-2212 and the spike trimer, illustrated with the high-resolution conformation 1 structure. Heavy chain and light chain CDR loops involved in direct interactions are shown in pink and cyan, respectively. Residues involved in hydrogen bond formation are shown in sticks with oxygen and nitrogen atoms colored red and blue, respectively. **e** Ribbon and stick diagrams showing the disruption of the key hydrogen bonds in E484 by the E484K mutation. **f** Surface diagrams showing mapping of key mutations in different variants and the positions of the mutated sites relative to the binding epitopes recognized by TAU-2212. The TAU-2212 heavy and light chain epitopes are colored yellow and white, respectively. The mutation sites within or outside the binding epitope of TAU-2212 are colored red and green, respectively. **g** Structural comparisons of the Fabs in conformations 1, 4, and 5. The alignments were performed by using the RBDs (in gray) and the variable regions of the bound Fabs. The bound Fabs of conformations 1, 4, and 5 are colored orange, black, and blue, respectively.

**Table 2 Cryo-EM data collection, refinement and validation statistics for spike–Fab2303 and spike–TAU-2212.**

| | S6P-Fab2303 | S2P–TAU-2212 | | | | |
|---|---|---|---|---|---|---|
| | | Conformation 1 | Conformation 2 | Conformation 3 | Conformation 4 | Conformation 5 |
| **Data collection and processing** | | | | | | |
| Magnification | 22500 | 29000 | 29000 | 29000 | 29000 | 29000 |
| Voltage (kV) | 300 | 300 | 300 | 300 | 300 | 300 |
| Electron exposure (e–/Å$^2$) | 50 | 50 | 50 | 50 | 50 | 50 |
| Defocus range (μm) | −1.5 to −2.8 | −1.5 to −2.0 | −1.5 to −2.0 | −1.5 to −2.0 | −1.5 to −2.0 | −1.5 to −2.0 |
| Pixel size (Å) | 1.25 | 0.97 | 0.97 | 0.97 | 0.97 | 0.97 |
| Symmetry imposed | C1 | C1 | C1 | C1 | C3 | C3 |
| Initial particle images (no.) | 546293 | 275642 | 275642 | 275642 | 275642 | 90569 |
| Final particle images (no.) | 38331 | 20339 | 15868 | 14193 | 72056 | 39788 |
| Map resolution (Å) | 4.5 | 4.7 | 7.3 | 6.4 | 3.3 | 6.1 |
| FSC threshold | 0.143 | 0.143 | 0.143 | 0.143 | 0.143 | 0.143 |
| Map resolution range (Å) | 4.5–30.0 | 4.7–30.0 | 7.3–30.0 | 6.4–30.0 | 3.3–6.7 | 6.1–14.8 |
| **Refinement** | | | | | | |
| Initial model used (PDB code) | | | | | 6XEY | |
| Model resolution (Å) | | | | | 3.25 | |
| FSC threshold | | | | | 0.143 | |
| Model resolution range (Å) | | | | | ∞−3.3 | |
| Map sharpening B factor (Å$^2$) | | | | | −135.0 | |
| Model composition | | | | | | |
| Non-hydrogen atoms | | | | | 33525 | |
| Protein residues | | | | | 4284 | |
| Ligands | | | | | 33 | |
| B factors (Å$^2$) | | | | | | |
| Protein | | | | | 24.30 | |
| Ligand | | | | | 35.12 | |
| R.m.s. deviations | | | | | | |
| Bond lengths (Å) | | | | | 0.005 | |
| Bond angles (°) | | | | | 0.977 | |
| Validation | | | | | | |
| MolProbity score | | | | | 1.97 | |
| Clashscore | | | | | 13.14 | |
| Poor rotamers (%) | | | | | 0.24 | |
| Ramachandran plot | | | | | | |
| Favored (%) | | | | | 95.14 | |
| Allowed (%) | | | | | 4.86 | |
| Disallowed (%) | | | | | 0.00 | |

Given the flexibility and dynamics of the interaction between TAU-2212 and the spike trimer, we examined binding of the Fab2212 variable regions to the adjacent RBDs in conformations 1–5. We noticed that the densities of bound Fabs2212 were considerably weaker in conformation 2 than in the other conformations, indicating that the "one up" - "two down" configuration may not be favorable for TAU-2212 binding (Fig. 5c). This possibility was also suggested by the pull-down results. To obtain the TAU-2212: spike complex, we applied more spike and a large portion of the spike protein was in the flowthrough which is likely to be in the RBD "up" conformations. Structural analysis of the bound Fabs indicated that the constant regions of the three bound Fabs in conformation 4 (three Fabs binding three "down" RBDs), are positioned at a small bending

angle (14.6°) relative to the variable regions that are aligned with the z axis (Fig. 5g). In addition, the three molecules are well separated (Fig. 5c) and the distances between the two C254s that form a disulfide bond and crosslink loops of the bound Fabs is ~21 Å (Supplementary Fig. 10), suggesting that the three bound Fabs belong to three different mAbs, rather than one mAb binding the trimer with both arms. The constant and variable domains of the bound Fabs in conformation 5 assume similar conformations as these in conformation 4, suggesting that the bound Fabs in conformation 5 belong to three different mAbs. In contrast to conformation 4, only two Fabs are present per trimer in conformation 1. The constant regions of the two bound Fabs in conformation 1 have a large bending angle (29.6°) and are joined at the distal end. In addition, the two C254s that crosslink loops of the bound Fabs have a reasonable distance of 4 Å (Supplementary Fig. 10), suggesting that in this case, the two bound Fabs belong to the same mAb (Fig. 5c and Supplementary Fig. 10). Notably, conformation 3 has the three bound Fabs in two distinct conformations. Two of the Fabs have similar bending angle as that observed in conformation 1, whereas one of the Fabs has similar bending angle as that observed in conformation 5, indicating that the binding of TAU-2212 in conformation 3 is in a mixed mode with two mAbs. Among these, one mAb contributes two Fabs, while one mAb provide only one Fab (Fig. 5c).

Structural analysis of the head-to-head spikes in conformation 5 reveal three mAbs that crosslink two spikes, with the two Fabs of each mAb in two linearly aligned 180° opposite positions (Fig. 5c and Supplementary Fig. 10c). As a result, the constant region of the bound Fab in conformation 5 has the smallest bending angle and is almost in a plane with the variable region. Given that most particles are in conformation 4, and no spike trimers were observed with all "down" RBDs and a single bound TAU-2212, we can deduce that binding of a single TAU-2212 mAb to the down RBD trimer triggers a conformational change that promotes additional Fab binding. With the binding of the first Fab, the second Fab of the bound mAb will be prone to binding the neighboring epitope. However, binding of both the Fabs in one mAb will cause bending in the bound Fabs as shown in conformation 2, which will reduce the stability of the interaction. Thus, the binding mode in conformation 1 could soon be replaced by the binding mode in conformation 4, whereas conformation 3 is likely an intermediate state between conformations 1 and 4. Taken together, mAb TAU-2212 adopts a unique binding mode to the spike. mAb TAU-2212 recognizes adjacent RBDs in a "down" conformation and crosslinks two spikes facing each other. However, a single Fab2212 barely binds the spike trimer, which is completely different from mAbs S2M11[37] and C144[38]. Like mAb TAU-2212, S2M11 and C144 IgGs could also cause the head-to-head spike linkage. Our result indicate that natural IgG can also crosslink spike and promote virus particle aggregation as the "bispecific" nanobody Fu2 does[39], although the RBD conformation and epitopes are different in these complexes.

## Discussion
Despite the relatively stable genome of SARS-CoV-2[40], the continuing spread of the global pandemic has been accompanied by the emergence of new variants with improved transmissibility and mutations that contribute to immune evasion[25,41–44]. With enhanced affinity for human cells and alterations made to vulnerable residues within the spike, these new variants can jeopardize both mAb therapies and vaccines. The Alpha, Beta, Gamma, Delta, and Omicron VOCs are of particular interest as they completely replaced the original Wuhan-Hu-1 strain in

subsequent "waves" of the pandemic. The first part of our study assessed the binding inhibition and neutralization of nine antibodies, previously isolated from Wuhan-Hu-1 SARS-CoV-2 infected individuals[9]. Consistent with other reports[45,46], we show that ACE2bs mAbs are more affected by viral mutations than mAbs that bind to regions outside the ACE2bs. While these are generally less potent against the original infecting virus, non-ACE2bs mAbs appear to have broader activity against emerging variants.

In the second part of our study, we investigated the mechanism of neutralization of two neutralizing receptor-blocking mAbs, TAU-2303 and TAU-2212, at the atomic level. The atomic structure of Fab2303:RBD confirmed that TAU-2303 binds a surface that is also bound by the ACE2 receptor, as we observed by functional assays. In contrast to other ACE2bs mAbs in our study, TAU-2303 remained active against the Delta VOC, likely due to the comparable modes of binding of ACE2 and TAU-2303 with respect to both epitope and angle of approach to the RBD. However, like the majority of mAbs that block receptor binding, TAU-2303 was less effective against the Beta and completely ineffective against Omicron variant. We next examined the atypical mAb, TAU-2212, which exhibits an unusual recognition flexibility type of binding involving five different conformations. TAU-2212 binds and crosslinks "down" RBDs by displaying exceptional conformational flexibility. According to the fitted structure of each conformation, the interfaces between the RBD and the variable region of the bound Fab2212 are consistent. We considered that the binding surface remains the same among the five conformations. The observation that TAU-2212 binds the spike complex in five different conformations suggests a flexibility of neutralization that is also achieved by stabilization of the spike trimer. The existence of a large number of TAU-2212 mAb crosslinked with head-to-head spikes suggests that TAU-2212 can potently crosslink and aggregate viral particles and thereby reduce the number of effective viral particles in the lung. Furthermore, the binding mode in conformations 1, 3, 4 or 5 would block any "up" conformation of the RBDs, thus, blocking receptor binding. In these ways, TAU-2212 exhibits ACE2bs-like properties while utilizing a distinct mechanism of action. Alternative strategies that broaden neutralization capacity can be deduced from the previously reported antibodies S2M11 and C144 which crosslink two "down" RBDs and have a similar mode of binding to that of TAU-2212 (Supplementary Fig. 11). However, unlike TAU-2212, both S2M11 and C144 can bind the spike trimer with the Fab alone. Out of the three antibodies, S2M11 has the largest binding interface, while the binding interfaces of TAU-2212 and C144 are smaller and more similar (Supplementary Fig. 11). Like TAU-2212, S2M11 mAb promotes a compact RBD head, whereas C144 binding promotes the opposite effect by forcing the RBD head into an open conformation (Supplementary Fig. 11b). The larger binding interface of S2M11 encompasses E484 and L452 residues, and therefore is expected to lose efficacy against the Beta, Gamma and Delta variants, whereas TAU-2212 effectively neutralizes Delta (Fig. 3 and Supplementary Fig. 11). Like other mAbs in its class, TAU-2212 also exhibited a complete loss of activity against the Beta, Gamma and Omicron VOCs.

To summarize, our study provides functional and atomic-level structural data on the interactions between naturally elicited antibodies and SARS-CoV-2 variants. Both TAU-2303 and TAU-2212 are potently neutralizing but arise through different B cell developmental programs. Neutralization by TAU-2212 is successful for most of the mutations with the exception of E484K. We therefore conclude that combining mAbs that can bind E484K, such as TAU-1109, −2303 or −2310 with TAU-2212 may be useful for broad spectrum anti-viral neutralization.

## Methods

**Cloning and mutagenesis of variant SARS-CoV-2 RBDs**. We used our previously reported wild type plasmid[9] as template for PCR mutagenesis designed to generate RBD constructs harboring single amino acid mutations. Pairs of overlapping DNA primers containing one or two base pair substitutions flanked by 20 bases on each side were designed and synthesized by Syntezza-Israel. PCR reactions were performed using KAPA HiFi HotStart ReadyMix (Roche) DNA polymerase. Each PCR reaction contained 10 μL KAPA HiFi HotStart ReadyMix, 0.5 μM of each primer, and 1 ng template DNA, with the sample volume adjusted to 20 μL with DNase/RNase free water (Bio-Lab). The PCR conditions were as follows: 95 °C for 3 min, 16 cycles of 98 °C for 20 s, and 72 °C for 90 s. Double and triple amino acid mutants were generated similarly with appropriate templates and primers.

**Expression and purification of soluble SARS-CoV-2 RBDs for ELISA**. Each construct was used to transiently transfect Expi293F cells (Thermo Fisher) using the ExpiFectamine 293 Transfection Kit (Thermo Fisher). Seven days post transfection, the cell supernatant was collected, filtered (0.22 μm), and incubated with $Ni^{2+}$-NTA agarose beads (GE Life Sciences) for 2 h at room temperature (RT). Proteins were eluted by 200 mM imidazole, buffer-exchanged to PBS ×1, aliquoted, and stored at −80 °C.

**ELISA**. High-binding 96 well ELISA plates (Corning #9018) were coated with 1 μg/mL RBD in PBS ×1 overnight at 4 °C. The following day, the coating was discarded, the wells were washed with "washing buffer" (PBS ×1 and 0.05% Tween20) and were blocked for 2 h at RT with 200 μL of "blocking buffer" (PBS ×1, 3% BSA (MP Biomedicals), 20 mM EDTA, and 0.05% Tween20 (Sigma)). Antibodies were added at a starting concentration of 4 μg/mL, and seven additional 4-fold dilutions in blocking buffer, and incubated for 1 h at RT. The plates were then washed 3 times with washing buffer before adding secondary, anti-human IgG (Jackson ImmunoResearch) antibody conjugated to horseradish peroxidase (HRP) diluted 1:5000 in blocking buffer, and incubation for 1 h at RT. Following four additional washes, 100 μL of TMB (Abcam) was added to each well and the absorbance at 650 nm was read after 20 min (BioTek 800 TS).

**Antibody inhibition of hACE2 binding to cell-expressed spike**. Expi293F cells were transfected with pcDNA 3.1 containing SΔC19 of wild type, Alpha, Beta, or Delta variants, using the ExpiFectamine 293 Transfection Kit (Thermo Fisher). The following day, the cells were harvested, centrifuged at $300 \times g$, and resuspended in FACS buffer (PBS ×1, 2% FBS and 2 mM EDTA). Next, the cells were aliquoted into a 24-well plate (Corning), so that each well contained $3 \times 10^6$ cells in 1 mL of FACS buffer. TAU antibodies or mGO53 were added to the appropriate wells at a concentration of 20 μg/mL with unlabeled hACE2 at a concentration of 1 μg/mL. The cells were then incubated for 30 min in an 8% $CO_2$ incubator with gentle shaking, transferred to FACS tubes, washed with FACS buffer, and incubated with biotinylated hACE2 for 20 min at 4 °C. Following an additional washing step, the cells were incubated with 0.5 μg of streptavidin-APC (Miltenyi Biotec, 130-106-792) and washed again. APC florescence was recorded using a CytoFLEX S4 (Beckman Coulter).

**Pseudo-particle preparation and neutralization assays**. SARS-CoV-2-spike pseudo-particles were obtained by co-transfecting Expi293F cells with pCMV delta R8.2, pLenti-GFP (Genecopoeia), and pcDNA 3.1 SΔC19 (Thermo Fisher) at a ratio of 1:2:1, respectively, according to the manufacturer's instructions. The supernatant was harvested 72 h post transfection, centrifuged at $1500 \times g$ for 10 min and passed through a 0.45 μm filter (LIFEGENE, Israel). The supernatant was then concentrated to 5% of its original volume using an Amicon Ultra with a 100 KDa cutoff at 16 °C (Merck Millipore). HEK-293 cells stably expressing hACE2 were seeded into 0.1% gelatin-coated 96-well plates (Greiner) at an initial density of $0.75 \times 10^5$ cells per well. The following day, concentrated pseudo-particles were incubated with serial dilutions of antibodies for 1 h at 37 °C and added to the 96 well plates. After 48 h, the cell medium was replaced with fresh DMEM medium excluding Phenol Red, and 24 h later, the 96-well plates were imaged by IncuCyte ZOOM (Essen BioScience). Cells were imaged with a 10× objective using the default IncuCyte software settings, which were used to calculate number of GFP-positive cells from four 488 nm-channel images in each well (data for each antibody was collected in triplicates). The number of GFP-positive cells was normalized and converted to a neutralization percentage. Pseudo-particles expressing the Alpha, Beta and Delta spikes were produced and tested similarly (Supplementary Table 1).

**Virus preparation and titer determination**. All work with SARS-CoV-2 was conducted under Biosafety Level-3 conditions at the University of California San Diego. SARS-CoV-2 isolates WA1 (USA-WA1/2020, NR-52281), Beta (B.1.351, hCoV-19/South Africa/KRISP-K005325/2020, NR-54009), Gamma (P.1, hCoV-19/Japan/TY7-503/2021, NR-54982), and Delta (B.1.617.2, hCoV-19/USA/PHC658/2021, NR-55611) were acquired from BEI Resources. Viral stocks originally isolated on VeroE6 were passaged once through primary human bronchial epithelial cells (NHBECs) differentiated at air-liquid interface (ALI) before expansion on VeroE6-TMPRSS2 (Sekisui XenoTech), referred to here as Vero-TMPRSS2. Variant Alpha

(B.1.1.7) was isolated on NHBECs at ALI from a nasopharyngeal swab obtained under UCSD IRB #200477 and expanded on Vero-TMPRSS2. The isolate has been deposited at BEI Resources as hCoV-19/USA/CA_UCSD_5574/2020, NR-54020. Variant Omicron (BA.1) was isolated and expanded on Vero-TMPRSS2 cells from a nasopharyngeal swab obtained under UCSD IRB #160524 with sequence deposited at GISAID (EPI_ISL_8186377). All viral stocks were verified by deep sequencing.

Virus titers were validated using a combination of fluorescent focus assay and tissue culture infectious dose $(TCID)_{50}$ assays on Vero-TMPRSS2 and Calu-3 (ATCC) monolayers. Serial dilutions of virus stocks in DMEM (Corning, #10-014-CV) were added to Vero-TMPRSS2 monolayers in 96-well plates and incubated for 1 h at 37 °C with rocking. For fluorescent focus assays, cells were overlaid with 1% carboxymethylcellulose in DMEM supplemented wth 2% FBS and 1x Pen/Strep. Plates were incubated at 37 °C in 5% $CO_2$ for 24 h depending on the assay and then fixed with a final concentration of 4.5% formaldehyde for at least 30 min at RT. For $TCID_{50}$ assays, Vero-TMPRSS2 or Calu-3 cells were infected as above and 100 μL DMEM or MEM with 2% FBS was subsequently added per well. Monolayers were observed for at least 4 days for appearance of CPE and then fixed as above and stained with antibody against SARS-CoV-2 Nucleocapsid (1 μg/mL). $TCID_{50}$ was calculated using the Reed-Muench method[47–49].

**Immunofluorescence imaging and analysis**. For viral nucleocapsid detection in Vero-TMPRSS2 cells by immunofluorescence, cells were washed twice with PBS ×1 and fixed in 4% formaldehyde for 30 min at RT. Fixed cells were washed with PBS ×1 and permeabilized for immunofluorescence using BD Cytofix/Cytoperm according to the manufacturer's protocol for fixed cells, and then stained for SARS-CoV-2 with a primary nucleocapsid antibody (1 μg/mL) (GeneTex GTX135357) and a secondary anti-rabbit AF594 antibody (ThermoFisher, A11037), The nuclei were counterstained with 1 μM Sytox Green. Infected cells from whole well scans were identified using the Incucyte S3 (Sartorius). Data were logged from the Incucyte analysis modules and graphed with GraphPad Prism 8.

**Expression and purification of soluble RBD, spike and antibodies for crystallization**. SARS-CoV-2 RBD (residues Arg319 to Lys529) was expressed by using the Bac-to-Bac Baculovirus System (Invitrogen). RBD containing the gp67 signal peptide and a C-terminal 6×His tag was inserted into pFastBac1 to form the plasmid pFastBac1-RBD. The plasmid was then transformed into DH10 Bac component cells. The recombinant bacmid was extracted and further transfected into Sf9 cells using Cellfectin II (Invitrogen, #10362100). The recombinant viruses were harvested from the transfected supernatant and amplified to generate high-titer virus stock. The viruses were then used to infect Hi5 cells for RBD expression. Secreted RBD in the supernatant was harvested and applied to cobalt agarose beads, then eluted with 300 mM imidazole, and further purified by using a Superdex 200 Increase 10/300 column (GE Healthcare) running in a buffer containing 20 mM HEPES at pH 8.0 and 150 mM NaCl.

The heavy chain and light chain of TAU-2303 or TAU-2212 were cloned separately into a pCMV vector and transiently transfected into HEK293F cells by using PEI at a ratio of 1:1. Supernatant was harvested 4 days after transfection. TAU-2212 was captured and purified using protein A beads (GE healthcare), eluted with 0.1 M glycine at pH 3.2, and then neutralized to pH 7.6.

For Fab preparation, the purified antibodies (TAU-2303 and TAU-2212) were digested by using the protease papain (Sigma, #P3125) with an IgG to papain ratio of 66:1 (w/w) for 3 h at 37 °C. The Fc domain and undigested antibodies were removed with protein A Sepharose (GE Healthcare) and the flowthrough was collected, and further purified by using a Superdex 200 Increase 10/300 column (GE Healthcare) running in a buffer containing 20 mM HEPES at pH 8.0 and 150 mM NaCl.

The extracellular domain of S protein (S-ECD) (1–1208 amino acids, Genebank ID: QHD43416.1) was cloned into the pCMV vector with six proline substitutions at residues 817, 892, 899, 942, 986, and 987 (S6P mutant) or two proline substitutions at residues 967 and 968 (S2P mutant)[50], a "GSAS" substitution at residues 682 to 685, and a C-terminal T4 fibritin trimerization motif followed by a StrepII tag (S6P mutant) or a Flag tag (S2P mutant). The S2P or S6P pCMV plasmids were used to transiently transfect HEK293F cells with polyethylenimine (PEI) (Polysciences, #24765). The recombinant S6P was affinity purified from the cell supernatant by using StrepTactin resin (IBA). The eluted material was further purified by size-exclusion chromatography using a Superose 6 10/300 column (GE Healthcare) running in 20 mM HEPES pH 8.0 and 150 mM NaCl. Supernatant containing S2P was collected 4 days after transfection, S2P was affinity purified by anti-Flag antibody beads and was eluted using 0.1 mg/mL 3×Flag peptide in 20 mM HEPES at pH 7.6 and 150 mM NaCl.

**Complex preparation and crystallization**. The Fab portion of TAU-2303 was mixed with RBD at a molar ratio of 1:2 at 4 °C, and the resultant complex was purified by size-exclusion chromatography with a Superdex 200 Increase 10/300 column running in 150 mM NaCl and 20 mM Tris-HCl, pH 8.0. The peak fractions containing the complex were collected and concentrated to 6.5 mg/mL for crystallization. The RBD-Fab2303 crystals were grown at 18 °C by using the hanging-drop vapor diffusion method with 1 μL protein mixed with 1 μL of reservoir

solution containing 0.2 M sodium tartrate dibasic dihydrate and 14% (w/v) polyethylene glycol 3350. Crystals were soaked in the reservoir solution supplemented with 15% glycerol, and flash-frozen in liquid nitrogen for data collection.

The Fab portion of TAU-2212 was concentrated to 7 mg/mL for crystallization. The crystal was grown at 16 °C in 26% (w/v) polyethylene glycol 3350, 0.2 M $(NH_4)_2SO_4$, pH 8.0. Crystals were soaked in the reservoir solution supplemented with 10% glycerol, and flash-frozen in liquid nitrogen for data collection.

**Data collection, structure determination, and refinement.** The X-ray diffraction data were collected at the beamlines BL18U (RBD-Fab2303) and BL02U (Fab2212) of the Shanghai Synchrotron Research Facility. The wavelength was 0.980 Å and the data collection temperature was 100 K. Data were processed and scaled with HKL2000[51]. The structure was determined by molecular replacement using PHASER[52]. Manual building and adjustments of the structures were performed in COOT[53]. The structure of RBD-Fab2303 was refined by using PHENIX[54]. Data processing showed that the crystal of Fab2212 belongs to the space group $P6_5$. However, the merohedral twinning appears to assign the crystal to the space group $P6_522$. The structure of Fab2212 was refined against the twinned data by using Refmac5[55] with the twinning operator k, h, -l and an estimated initial twin ratio of 0.52 to 0.48 between the two domains. The final refined twin ratio between the two domains were 0.60 to 0.40. Data collection and refinement statistics are listed in Table 1. The Ramachandran statistics are as follows: 97.24% favored and 2.76% allowed for RBD-Fab2303; 95.26% favored, 4.5% allowed and 0.24%outlier for Fab2212. In Fab2212 refinement, values in $R_{work}/R_{free}$ and R.m.s. deviations columns were obtained from refmac5 job output. Structural analysis of antibody–antigen contacts were assessed through CCP4i[56] (Supplementary Tables 2,3). All structural representations were prepared through the use of the UCSF Chimera and ChimeraX[57,58].

**CryoEM sample preparation, data acquisition, and processing.** Fab2303 was mixed with the purified S6P trimer (2:1 molar ratio Fab per protomer) to form the Fab-S complex at a concentration of 2 mg/mL. The mixture was incubated on ice for 30 min. Next, 3 μL of the mixture was applied to glow-discharge holey carbon grids (Quantifoil, Cu 200 mesh, R1.2/1.3). The grid was blotted for 4.5 s in 100% humidity before being flash-frozen in liquid ethane by using a Vitrobot Mark IV (Thermo Fisher). For TAU-2212-S2P complex preparation, S2P and TAU-2212 were incubated for 30 min on ice at a molar ratio of 1:2. The antibody-spike complex was purified by protein A beads. The neutralized elution was further purified by anti-flag antibody beads. The eluted complex was crosslinked by 0.125% glutaraldehyde for 20 min. The crosslinked sample was purified by size-exclusion chromatography with a Superose 6 10/300 column. The target fraction was collected and concentrated to 0.8 mg/mL for cryoEM grid preparation by using similar conditions to those used for S6P and Fab2303.

Image data of the S6P-Fab2303 and S2P-mAb-TAU2212 complexes were collected on a Titan Krios electron microscope (FEI Company) equipped with a field emission gun operated at 300 kV and a Gatan K3 Summit camera. Images of S6P-Fab2303 were recorded at a defocus range of −1.5 to −2.8 μm, with a pixel size of 1.25 Å. A total dose of ~ 50 electrons per Å$^2$ was used. SerialEM was used for the data collection. A total of 2599 movie stacks for S6P-Fab 2303 and 4675 movie stacks for S2P-TAU2212 were collected. The frames in each movie stack were aligned, summed, and 2× binned using Motion Cor2[59]. The CTF parameters of the micrographs were determined by Gctf with local defocus variations taken into consideration[60]. For the S6P-Fab2303 complex, a total of 546293 particles were boxed by using Gautomatch, and were then subjected to 2D classification using RELION[61]. The cryoEM structure of the closed state SARS-CoV-2 spike (PDB accession number: 6VXX[62]) was low-pass filtered to 40 Å and used as the initial model. A total of 38331 particles were selected for the final 3D refinement without imposing any symmetry, which yielded a cryoEM map at a resolution of 4.5 Å (Table 2).

For the S2P-mAb-TAU2212 complex, data was collected at defocus −1.5 to −2.0 μm, with pixel size of 0.97 Å. 90569 particles were picked by Gautomatch and then were subjected to 2D classifications with RELION 3.0. The 2D classification results showed two major classes, one with a single spike and the other with two head-to-head crosslinked spikes. Particles in the two classes were split for separated 3D classifications in RELION 3.0. For the head-to-head crosslinked spikes, 39788 particles were selected for 3D refinement with D3 symmetry imposed, which resulted in a cryoEM map at a resolution of 9.36 Å. To improve the reconstruction, the head-to-head spike particles were split into two particles each with one spike and three bound Fabs. The split particles were subjected to local refinements, which resulted in a cryoEM map at a resolution of 7.3 Å. The resolution of the map with a tight mask generated by cryoSPARC was improved to 6.1 Å (Supplementary Fig. 6).

Particles in classes with a single spike were selected by Gautomatch and subjected to 2D classification. Selected particles were subjected to 3D classification with a S trimer cryoEM map as the reference (PDB accession number: 6XEY[63]). This produced four distinct conformations: conformation 1 has two bound Fabs; conformation 2 has only one bound Fab; conformation 3 has two Fabs from one mAb and one Fab from the other mAb; and conformation 4 is 3-fold symmetric and has three bound Fabs. Particles from each conformation were selected and used for refinement against an RBD "all-down" spike that was low passed to 60 Å.

For conformation 4, the final refinements were performed with 72056 selected particles and C3 symmetry imposed. The resolution of the final reconstructed map is 3.5 Å. We applied the refined half maps from Relion to cryoSPARC for local resolution estimation and with the cryoSPARC generated auto-tighten mask and local filter tool, the resolution of the post processed map was improved to 3.3 Å. Side chain densities are clearly visible in most of the reconstructed map (Supplementary Fig. 8). For conformations 1, 2 and 3, the final refinements were performed with 20339, 15868 and 14193 selected particles, respectively, and without imposing any symmetry. The refinements resulted in a cryoEM map at a resolution of 5.5 Å for conformation 1, a cryoEM map at a resolution of 7.8 Å for conformation 2, and a cryoEM map at a resolution of 6.5 Å for conformation 3. The processing of the cryoEM map for conformation 1 was equivalent to comformation 4 by cryoSPARC, and the resolution was improved to 4.7 Å. To keep the Fab features in conformation 2 and 3, we used looser masks in cryoSPARC Local Filter jobs, and the final resolution were 7.3 Å and 6.4 Å, respectively.

All of the refined density maps were applied with a negative B-factor and corrected for the modulation transfer function (MTF) of the detector. The reported resolution is based on the gold-standard Fourier shell correlation (FSC) 0.143 criterion[64,65] (Supplementary Fig. 7).

**Statistics and reproducibility.** All graphs were generated and SD calculated using Prism versions 8 and 9.

**Reporting Summary.** Further information on research design is available in the Nature Research Reporting Summary linked to this article.

## Data availability

The atomic coordinates and EM maps have been deposited into the Protein Data Bank (http://www.pdb.org) and the EM Data Bank, respectively: Fab2303-RBD complex (PDB: 7WBZ), Fab2303-S complex (EMD: 32411), Fab2212 (PDB: 7WC0), mAb2212-S complex in conformation 1 (EMD: 32416), conformation 2 (EMD: 32417), conformation 3 (EMD: 32418), conformation 4 (EMD: 32421, PDB: 7WCD), conformation 5 with head-to-head spikes (EMD: 32420) and conformation 5 with single spike (EMD: 32419).

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

## Acknowledgements

We thank the members of the Xiang and Freund labs for fruitful discussions and assistance. We thank Noam Ben-Shalom and the Blavatnik Center for Drug Discovery in Tel Aviv University for help with SPR measurmints. NTF is funded by the ISF grant number #1422/18. MGT and NTF are funded by KillCorona ISF grant number #3711/20. YX is funded by the Spring Breeze Fund of Tsinghua University, the National Natural Science Foundation of China (grants: 31925023, 21827810, 31861143027), the Ministry of Science and Technology of China (grant 2021YFA1300200), the Beijing Frontier Research Center for Biological Structure, and the Beijing Advanced Innovation Center for Structure Biology. AFC is supported by NIH grant (K08 AI130381) and AFC is supported by Career Award for Medical Scientists from the Burroughs Wellcome Fund. MD is funded by the BARD grant #IS-5270-20R and ISF grant #401/18. BAC is supported by a United States-Israel Binational Science Foundation grant. The following reagent was deposited by the Centers for Disease Control and Prevention and obtained through BEI Resources, NIAID, NIH: SARS-Related Coronavirus 2, Isolate USA-WA1/2020, NR-52281. The following reagent was obtained through BEI Resources, NIAID, NIH: SARS-Related Coronavirus 2, Isolate hCoV-19/South Africa/KRISP-K005325/2020, NR-54009, contributed by Alex Sigal and Tulio de Oliveira. The following reagent was obtained through BEI Resources, NIAID, NIH: SARS-Related Coronavirus 2, Isolate hCoV-19/Japan/TY7-503/2021 (Brazil P.1), NR-54982, contributed by National Institute of Infectious Diseases. The following reagent was obtained through BEI Resources, NIAID, NIH: SARS-Related Coronavirus 2, Isolate hCoV-19/USA/PHC658/2021 (Lineage B.1.617.2; Delta Variant), NR-55611, contributed by Dr. Richard Webby and Dr. Anami Patel. We thank Dr. Louise Laurent for providing the B.1.1.7 clinical sample for viral isolation and the UC San Diego Center for Advanced Laboratory Medicine Microbiology Laboratory for providing the BA.1 clinical sample. We thank and UC San Diego EXCITE for SARS-CoV-2 genome sequencing.

## Author contributions

M.M. planned and performed the biochemical experiments, analysed data, prepared the figures, and wrote the manuscript together with N.T.F., Y.X., and B.A.C. R.L. and B.M. performed crystal and cryoEM structure determination and analysis, prepared the figures, and wrote the manuscript together with N.T.F., Y.X., and B.A.C. M.W., J.A., M.D. and M.G.T. performed the pseudo-viral assays. J.C.L. performed cell line maintenance and plate setup for BSL-3 assays. A.E.C. and A.F.C. conducted virus generation and titering. S.L.L. and B.A.C. designed and performed BSL-3 assays. Y.X., N.T.F., and B.A.C. planned and supervised the experiments, analyzed the data and wrote the manuscript.

## Competing interests

The authors declare no competing interests.

## Ethics statement

All the authors meet the authorship criteria, and gave their consent to be listed as authors on this manuscript.
