## [Peer Review File · Communications Biology]

Reviewers' comments:

Reviewer #1 (Remarks to the Author):

The authors present here a panel of SARS-CoV-2 specific antibodies that have been assayed for their ability to recognise and neutralise current variants of concern (VoC) in vitro. Two of these (TAU-1109 and TAU-2310) appear to be capable of broad neutralisation of VoC. Structural investigation using cryo-EM was performed on antibodies TAU-2303 and TAU-2212 and revealed firstly how TAU-2303 maintains recognition of the SARS-CoV-2 delta variant but is sensitive to beta and omicron variants. Secondly it was shown that TAU-2212 binds a quaternary epitope on the spike trimer and can bind in a number of stoichiometries, which may differ by whether single mAbs recognise multiple RBDs on the same spike, or if different sites on the trimer are bound by separate mAbs.

While antibodies recognising quaternary epitopes are interesting, this is not the first described, as acknowledged in the manuscript. The structural analysis of the differing 3D classes is engaging and gives some insight into how antibody recognition and neutralisation of Spike may differ depending upon the concentration of an antibody. Conformation 4 potentially representing a situation with the highest number of (full length) antibodies per trimer.

The work is performed well, and a large amount of data is presented. However, in order to understand the basis of neutralisation of VoC at an "atomic level" and "reveal sites with a lower tendency to variation" as stated in the introduction, it might have been more illuminating to include one of the antibodies in the panel (e.g. TAU-2310) that maintained recognition across all VoCs in their structural analysis.

Suggestions

1. Line 167: Structural basis of neutralization of the ACE2bs mAb TAU-2303 (figure 4a). Therefore, mAb TAU-2303 can be categorized as a CoV2130-165 type mAb, that belongs to Class 1 RBD binding mAbs, and binds only one RBD subunit amongst 166 the three available subunits of the trimer

It appears in the methods section that the complex for spike-TAU-2303 was set up with only a 2-fold molar excess of fAb to spike trimer, so full occupancy of the trimer could not be achieved. While I agree some variation in stoichiometry would be expected if binding of multiple sites was possible, have the authors ruled out that at complete saturation TAU-2303 cannot bind multiple RBDs on a single spike? Please include an explanation if so or re-word this statement.

2. Line 173: Structural basis of neutralization of the ACE2bs mAb TAU-2303. The contact residues mediate the formation of a large number of hydrogen bonds (Extended Data Table S3).

Could the authors please include the number of hydrogen bonds instead of a "large number" here.

3. Line 233: mAb TAU-2212 blocks ACE2 binding through conformational dynamics. The final model of Fab2212 contains 418 residues, while residues 136-148 and 192-201 of heavy chain are not visible in the map (Fig. 5a).

Please highlight the missing sections described here on figure 5a as they are not easily visible in the current figure.

4. Extended Data Figure 7:

Local resolutions from Resmap are quite hard to interpret here. Would it be possible to replace these with local resolutions estimated using relion/cryosparc implementations?

5. Extended Data Figure 7, 8:

While the RBD density for conformation 4 seems to be well defined in extended data 8, it is hard to

judge the local resolution of the RBD-antibody region in extended data figure 7. As the interface of TAU-2212 with the RBDs is discussed in detail from line 273, it would be useful to know the resolution of this area when considering the interactions described. Could the authors please show a zoom of this region for conformation 4 in extended data figure 7 to better indicate the resolution here.

6. Line 290: mAb TAU-2212 blocks ACE2 binding through conformational dynamics. Substitutions S371L and S373P that were reported in the new Omicron variant, also lie within the TAU-2212 interface. To test the effect of these mutations on TAU-2212 binding, we introduced a S373G mutation.

Could the authors please clarify this strategy, it seems unusual to test the potential impact of a serine to proline substitution at a residue by introducing a glycine instead?

7. Extended figure 10C:

When describing whether the fAbs in conformations 1, 4 and 5 originate from a single mAb or separate mAbs, it might be useful to include a simple schematic (or statement) alongside the current image for each of the 3 scenarios shown to make it easy for the reader.

8. Line 327: The nearby Fab from the same mAb may take the advantage to bind at first. However, the 327 distortion and bending of the bound Fabs from the same mAb make an unstable binding to spike.

This sentence may benefit from rewording to improve clarity.

9. Line 331: Taken together, mAb TAU-2212 adopts a unique binding mode to spike.

Could the authors please expand this sentence to better define the unique aspects of TAU-2212 binding. It is mentioned in the discussion that SM211 and C144 also recognise overlapping quaternary epitopes to TAU-2212, while the binding stoichiometries seen in the Cryo-EM presumably result from this being carried out using full length mAb rather than fAbs in previous studies. The head-to-head double spike orientation seen in conformation 5 has been published with spikes linked by a "bispecific" nanobody (Hanke et al, Nat comms 2022. <https://doi.org/10.1038/s41467-021-27610-z>).

10. Line 354: Discussion. We next examined the atypical mAb, TAU- 2212, which exhibits an unusual recognition flexibility type of binding involving five different possible conformations. TAU-2212 binds and crosslinks "down" RBDs by displaying an exceptional conformational flexibility

Could the authors please make explicit here that the epitope of TAU-2212 remains the same across all conformations, while positioning and stoichiometry of the antibody is variable.

11. Line 359: Discussion. The existence of a large number of TAU-2212 mAb crosslinked with head-to-head 359 spikes suggests that TAU-2212 can potently crosslink and aggregate viral particles

As a supporting reference for this hypothesis, the authors could cite that aggregation of SARS-CoV-2 virus by negative stain EM was shown for the nanobody which gave a similar head-to-head spike structure as in conformation 5 (Hanke et al, Nat comms 2022. <https://doi.org/10.1038/s41467-021-27610-z>).

12. Figure 4b:

Where the surface representations are rotated from the ribbon diagrams on the left, please add in the approximate amount (degrees) and direction of rotation to help the reader orient these structures (as done in 4e).

13. Extended Data Figure 1:

Could the authors please include IC50 values and errors in either bar chart format or as a table, as in the current format it isn't possible to get any real quantitative idea of binding from these curves.

14. Extended Data Figure 6:

Please indicate the software used for each step on the figure for easy reference.

15. Extended Data Figure 11b legend:

Should this read TAU-2303 or TAU-2212?

16. Extended Data Table 3:

Please indicate how these contacts were determined (e.g PDBePISA, CCP4 Contact and any relevant parameters or cutoffs applied).

17. Extended Data Table 4:

Currently this reads as having 94.81 % of residues in the disallowed region for Ramachandran. Please could the authors correct this.

Reviewer #2 (Remarks to the Author):

The manuscript describes the binding of two neutralising Abs, namely Tau-2303 and -2212, to the RBD and to the S ectodomain of the SARS-CoV-2 Wuhan-Hu-1 and specific VOC. The results show quite convincingly, in my opinion, that Abs recognising the same RBD surface area as ACE2, and thus inhibiting ACE2 binding, are more susceptible to loss of affinity due to mutations of that region, which is highly targeted in the VOC.

Tau-2303 binds one RBD through the 1 Fab region and is less effective against beta and completely inactive against omicron. Tau-2212 has a very interesting, flexible recognition mode of action, showing 5 different conformations in complex with the S ectodomain, and a preference for a prefusion conformation with open RBDs, if I understood correctly, which may not be stable (prone to shedding) when S is not anchored to a viral/cell surface (the authors may want to comment on this). Still, because of RBD mutations also Tau-2212 binding is inactivated by omicron.

I found the work quite interesting and well presented, and for this reason I suggest its publication in the current form. As an important note, I am not an expert in the experimental methods used in this work and thus cannot comment on any of the technical aspects, leaving this task to reviewers who are experts in those fields.

Reviewer #3 (Remarks to the Author):

In this study the authors claimed that RBD specific ACE2 binding site targeting antibodies are more sensitive to variants of concern (VOCs), whereas non-ACE2bs RBD targeting antibodies are not as sensitive to VOCs. Using a set of patient-derived monoclonal antibodies the authors demonstrate this through ELISA binding assays, cell-based flow cytometry assays, and neutralization assays. They further structurally characterize two of their nine antibodies, one of which is an ACE2bs antibody that is somewhat variant resistant, and the other, a unique antibody that targets a quaternary epitope devised of the junctions on two adjacent RBDs in the context of the Spike trimer. Altogether this study adds to the multitude of different RBD specific monoclonals that have been described and characterized, suggesting that though the RBD is a relatively small part of the Spike protein there seems to be many different modes of binding that can result in neutralization, particularly by both directly and indirectly affecting ACE-2 binding.

However, the following points should be addressed.

1. It would be informative if the authors ran neutralization tests for the non-ACE2bs mAbs TAU-1109 and -2310 against other sarbecoviruses such as SARS-CoV-1, WIV1-CoV, and SHC014-CoV to see if their breadth extends beyond just SARS-CoV-2. Since these antibodies appear to be the most variant resistant, therefore the most clinically useful, structurally they are probably the most interesting targets, and there has been no further characterization of these class of antibodies in the paper. Have there been attempts to solve the structure of at least one of these antibodies in complex with Spike or RBD?

2. The authors use Pisa modelling in figure 4d to show how the substitutions found in the Beta VOC would affect the binding of TAU-2303 to the RBD. Since the changes in binding energy are quite small it's uncertain how meaningful these measurements are. It should be possible to show this how TAU-2303 accommodates these substitutions with an actual structure of the Beta RBD and TAU2303 complex. SPR or some other binding affinity measurement, might also help to pinpoint differences in binding affinity of TAU2303 with beta and other VOCs.

3. I am not sure how the data supports this statement: "TAU-2212 acts by binding and stabilizing the SARS-CoV-2 spike trimer in a "down" orientation and prevents conformational change to "up" RBD which is required for spike:ACE2 interactions." It might be better to elaborate on what "small" means in terms of the rotational change towards the trimer axis and whether this is a big contributor to stabilization? The authors also don't explore the context in which the RBD loops are disordered - is that just the case in the RBD when expressed alone, or is that true in the full-length trimer, and if so, are they ordered only upon ACE2 binding and otherwise disordered in all confirmations?

. I think a way to investigate this mechanism would be an ELISA where you look for binding to a class 4 epitope such as CR3022 to the Spike, in the presence of binding of TAU-2122. If the Trimer is locked in the down conformation, then CR3022 can't bind since its epitope is only exposed when the RBD is in the up conformation.

Response to the Reviewers:

We thank the reviewers for taking the time to evaluate our manuscript for their constructive feedback. Below please find a detailed point-to-point responses to their suggestions.

Color code:

Reviewer's text

Author's response

Manuscript text

Changes in the text

Reviewer #1 (Remarks to the Author):

The authors present here a panel of SARS-CoV-2 specific antibodies that have been assayed for their ability to recognise and neutralise current variants of concern (VoC) in vitro. Two of these (TAU-1109 and TAU-2310) appear to be capable of broad neutralisation of VoC. Structural investigation using cryo-EM was performed on antibodies TAU-2303 and TAU-2212 and revealed firstly how TAU-2303 maintains recognition of the SARS-CoV-2 delta variant but is sensitive to beta and omicron variants. Secondly it was shown that TAU-2212 binds a quaternary epitope on the spike trimer and can bind in a number of stoichiometries, which may differ by whether single mAbs recognise multiple RBDs on the same spike, or if different sites on the trimer are bound by separate mAbs. While antibodies recognising quaternary epitopes are interesting, this is not the first described, as acknowledged in the manuscript. The structural analysis of the differing 3D classes is engaging and gives some insight into how antibody recognition and neutralisation of Spike may differ depending upon the concentration of an antibody. Conformation 4 potentially representing a situation with the highest number of (full length) antibodies per trimer. The work is performed well, and a large amount of data is presented. However, in order to understand the basis of neutralisation of VoC at an “atomic level” and “reveal sites with a lower tendency to variation” as stated in the introduction, it might have been more illuminating to include one of the antibodies in the panel (e.g. TAU-2310) that maintained recognition across all VoCs in their structural analysis.

We thank the Reviewer for hers/his feedback and for pointing out that our study is well preformed.

Suggestions

1. Line 167: Structural basis of neutralization of the ACE2bs mAb TAU-2303 (figure 4a). Therefore, mAb TAU-2303 can be categorized as a CoV2130-165 type mAb, that belongs to Class 1 RBD binding mAbs, and binds only one RBD subunit amongst 166 the three available subunits of the trimer. It appears in the methods section that the complex for spike-TAU-2303 was set up with only a 2-fold molar excess of fAb to spike trimer, so full occupancy of the trimer could not be achieved. While I agree some variation in stoichiometry would be expected if binding of multiple sites was possible, have the authors ruled out that at complete saturation TAU-2303 cannot bind multiple RBDs on a single spike? Please include an explanation if so or re-word this statement.

We thank the Reviewer for bringing this up. Fab2303 and the S protomer were in a molar ratio of 2 (Fab):1(S protomer). Thus, the molar ratio between Fab2303 and the trimer should be 6(Fabs):1(S trimer), indicating that the Fab is abundant. We only observed one Fab2303 binding the protruding RBD of the spike. We have clarified this in the method.

2. Line 173: Structural basis of neutralization of the ACE2bs mAb TAU-2303. “The contact residues mediate the formation of a large number of hydrogen bonds (Extended Data Table S3).” Could the authors please include the number of hydrogen bonds instead of a “large number” here.

We thank the Reviewer for his comment. We included the following clarification in the text, line 185:

May 2022

“The contact residues mediate the formation of 23 hydrogen bonds.”

3. Line 233: mAb TAU-2212 blocks ACE2 binding through conformational dynamics. The final model of Fab2212 contains 418 residues, while residues 136-148 and 192-201 of heavy chain are not visible in the map (Fig. 5a). Please highlight the missing sections described here on figure 5a as they are not easily visible in the current figure.

We thank the Reviewer for his comment. We have highlighted the missing residues in Figure 5a and Extended Data Figure 4a.

4. Extended Data Figure 7: Local resolutions from Resmap are quite hard to interpret here. Would it be possible to replace these with local resolutions estimated using relion/cryosparc implementations?

We thank the Reviewer for this comment. We have updated the local resolution map generated by cryosparc. We applied the refined half maps from Relion to cryoSPARC for local resolution estimation and with the cryoSPARC generated auto-tighten mask and local filter tool, the resolution of post processed map was improved. We, thus, updated all the resolution data according to cryoSPARC results.

5. Extended Data Figure 7, 8: While the RBD density for conformation 4 seems to be well defined in extended data 8, it is hard to judge the local resolution of the RBD-antibody region in extended data figure 7. As the interface of TAU-2212 with the RBDs is discussed in detail from line 273, it would be useful to know the resolution of this area when considering the interactions described. Could the authors please show a zoom of this region for conformation 4 in extended data figure 7 to better indicate the resolution here.

We have added a new panel in Extended data Figure 7 showing three slices of local resolution map including the RBD-antibody interface region.

6. Line 290: “mAb TAU-2212 blocks ACE2 binding through conformational dynamics. “ S371L and S373P that were reported in the new Omicron variant, also lie within the TAU-2212 interface. To test the effect of these mutations on TAU-2212 binding, we introduced a S373G mutation.” Could the authors please clarify this strategy, it seems unusual to test the potential impact of a serine to proline substitution at a residue by introducing a glycine instead?

We thank the Reviewer for bringing this up. We intended to disrupt the hydrogen bond mediated by the hydroxyl group of Serine 373 and at the same time to remove effects of the side chain, which may establish new Val del Wall interactions and interfere with our judgement on the role of the hydrogen bond. Thus, we made the mutant S373G. Our result indicated that the hydrogen bond is not necessary for the RBD-antibody interaction. Next we performed a pull down experiment to test the interaction between mutant S373P and TAU-2212. The results showed that TAU-2212 can still bind spike with the S373P mutation, which further supports that both the hydrogen bond and Van Der Waals force mediated by S373 side chain do not play a major role in TAU-2212 binding spike. To clarify this, we added a gel image with the S373P mutant (now an additional panel Extended Data Fig. 5c), and we have added the following text in Line 333:

Substitutions S371L and S373P that are present in the new Omicron variant, also lie within the TAU-2212 interface. To test the effect of S373P mutation on TAU-2212 binding, we first introduced a S373G mutation to disrupt the hydrogen bond mediated by the hydroxyl group of serine 373. The results showed that the S373 mediated hydrogen bond did not affect the binding of TAU-2212 (Extended Data Fig. 5b). Next, we performed a pull down experiment to test the interaction between mutant S373P and TAU-2212 (Extended Data Fig. 5c). The results showed that TAU-2212 can still bind spike with the S373P mutation, which further supports that both the hydrogen bond and Van Der Waals force mediated by S373 side chain do not play a major role in TAU-2212 binding spike. Taking the effect of all the mutations together, it is therefore not surprising that TAU-2212 does not neutralize Omicron.

May 2022

7. Extended figure 10C: When describing whether the fAbs in conformations 1, 4 and 5 originate from a single mAb or separate mAbs, it might be useful to include a simple schematic (or statement) alongside the current image for each of the 3 scenarios shown to make it easy for the reader.

We have updated the Figure, according to the Reviewer's suggestion, with a simple schematic diagram showing the 3 scenarios.

8. Line 327: The nearby Fab from the same mAb may take the advantage to bind at first. However, the 327 distortion and bending of the bound Fabs from the same mAb make an unstable binding to spike. This sentence may benefit from rewording to improve clarity.

We thank the Reviewer for this point, and we apologize that the original text was unclear. We have changed the sentence in the resubmitted version, Line 378:

“With one Fab of the mAb binding, the other Fab of the bound mAb will be prone to binding the neighboring epitope. However, binding of both the Fabs in one mAb will cause bending in the bound Fabs as shown in conformation 2, which lead to the binding unstable.”

9. Line 331: Taken together, mAb TAU-2212 adopts a unique binding mode to spike. Could the authors please expand this sentence to better define the unique aspects of TAU-2212 binding. It is mentioned in the discussion that SM211 and C144 also recognise overlapping quaternary epitopes to TAU-2212, while the binding stoichiometries seen in the Cryo-EM presumably result from this being carried out using full length mAb rather than fAbs in previous studies. The head-to-head double spike orientation seen in conformation 5 has been published with spikes linked by a “bispecific” nanobody (Hanke et al, Nat comms 2022. <https://doi.org/10.1038/s41467-021-27610-z>).

We thank the Reviewer for this comment. Although the epitope is not unique, we focused on the binding mode of natural IgG that share the common epitope. In addition, Fab2212 cannot bind the S, which makes it completely different to the binding mode of S2M11 and C144. Both antibodies S2M11 and C144 are supposed to have similar binding mode in full length. Since this bispecific nanobody did not reflect the real IgG generated by immune system, we have compared our results with the nanobody. We added the relevant citation and to clarify, we added in the text Line 384:

“Taken together, mAb TAU-2212 adopts a unique binding mode to the spike. mAb TAU-2212 recognizes adjacent RBDs in “down” conformation and crosslinks two spikes facing each other. However, single Fab2212 barely binds spike, which is completely different from S2M11 and C144. It is supposed that S2M11 and C144 in full length will also cause the head-to-head spike linkage. Although head-to-head S spike conformation can be mediated by a “bispecific” nanobody, the RBD conformation and epitope are definitely different. Our result indicate that natural IgG can also crosslink spikes and be capable to promote virus particles aggregation as the “bispecific” nanobody Fu2 does (Hanke et al., 2022), although the RBD conformation and epitope are definitely different in these complexes.”

10. Line 354: Discussion. We next examined the atypical mAb, TAU- 2212, which exhibits an unusual recognition flexibility type of binding involving five different possible conformations. TAU-2212 binds and crosslinks “down” RBDs by displaying an exceptional conformational flexibility. Could the authors please make explicit here that the epitope of TAU-2212 remains the same across all conformations, while positioning and stoichiometry of the antibody is variable.

We thank the Reviewer for this comment. We therefore have added sentence in Line 431:

“According to the map of each conformation fitting with spike and Fab2212 models, structures around the interface of RBD and variable region of Fab2212 are consistent. We considered that the binding surface remains the same among the five conformations.”

May 2022

11. Line 359: Discussion. The existence of a large number of TAU-2212 mAb crosslinked with head-to-head 359 spikes suggests that TAU-2212 can potently crosslink and aggregate viral particles. As a supporting reference for this hypothesis, the authors could cite that aggregation of SARS-CoV-2 virus by negative stain EM was shown for the nanobody which gave a similar head-to-head spike structure as in conformation 5 (Hanke et al, Nat comms 2022. <https://doi.org/10.1038/s41467-021-27610-z>).

We thank the Reviewer for this advice, we have added this citation to support our hypothesis.

12. Figure 4b: Where the surface representations are rotated from the ribbon diagrams on the left, please add in the approximate amount (degrees) and direction of rotation to help the reader orient these structures (as done in 4e).

We have added the degree and direction in Fig. 4b.

13. Extended Data Figure 1: Could the authors please include IC50 values and errors in either bar chart format or as a table, as in the current format it isn't possible to get any real quantitative idea of binding from these curves.

We are sorry for the confusion. Extended Data Figure 1 depicts a representative binding data by ELISA (and not inhibition data), that are related to the quantified area under the curve (AUC) values that are presented in the main Figure 1a. Since the figure does not present inhibition data, but only binding, it would not be straight forward to extract IC50 values from this. As to the requested error bars, the presented ELISA is a representative experiment that was repeated at least three times, however performing error bar analysis on different ELISA repetitions would be incorrect. The mode of presentation shown in Figure 1 is used in many similar publication, such as in Turner and colleagues¹ (Figure 3), in Schmitz and colleagues² (Figure 1) and in Zhou and colleagues³ (Figure 1). The quantitative binding measurements by SPR were performed on the wt RBD in our previous publication (Mor *et al.* PloS Pathogens⁴), where affinity values for all the mAbs presented can be found. Moreover, in the revised manuscript we include additional SPR data for mAb TAU-2303 for all the VOCs.

14. Extended Data Figure 6: Please indicate the software used for each step on the figure for easy reference.

We thank the Reviewer for bringing this up. The data procession was all performed by using relion3.0.8. and cryoSPACR. We have indicated the software used in the figure legend.

15. Extended Data Figure 11b legend: Should this read TAU-2303 or TAU-2212?

We apologize this was not clear. Extended Data Figure 11 showing the epitopes of antibodies S2M11 and C144, which have a similar binding epitope as that of TAU-2212, with panel A showing the epitope of S2M11 and panel B showing the epitope of C144. We have added a title in the figure to clarify this.

16. Extended Data Table 3: Please indicate how these contacts were determined (e.g PDBePISA, CCP4 Contact and any relevant parameters or cutoffs applied).

We thank the Reviewer for the question. As mentioned in the end of the Table 3 and 5, contact residues were determined by CCP4 The distance cutoff used is 4.0 Å for determining the contact residues and the distance and angle cutoff is 4.0 Å and 90-270 degree, respectively, for determining the hydrogen bonds with PISA.

17. Extended Data Table 4: Currently this reads as having 94.81 % of residues in the disallowed region for Ramachandran. Please could the authors correct this.

We are sorry for the mistake and have corrected this in the corresponding Table.

May 2022

Reviewer #2 (Remarks to the Author):

The manuscript describes the binding of two neutralising Abs, namely Tau-2303 and -2212, to the RBD and to the S ectodomain of the SARS-CoV-2 Wuhan-Hu-1 and specific VOC. The results show quite convincingly, in my opinion, that Abs recognising the same RBD surface area as ACE2, and thus inhibiting ACE2 binding, are more susceptible to loss of affinity due to mutations of that region, which is highly targeted in the VOC.

Tau-2303 binds one RBD through the 1 Fab region and is less effective against beta a completely inactive against omicron. Tau-2212 has a very interesting, flexible recognition mode of action, showing 5 different conformations in complex with the S ectodomain, and a preference for a prefusion conformation with open RBDs, if I understood correctly, which may not be stable (prone to shedding) when S is not anchored to a viral/cell surface (the authors may want to comment on this). Still, because of RBD mutations also Tau-2212 binding is inactivated by omicron.

I found the work quite interesting and well presented, and for this reason I suggest its publication in the current form. As an important note, I am not an expert in the experimental methods used in this work and thus cannot comment on any of the technical aspects, leaving this task to reviewers who are experts in those fields.

We thank the Reviewer for the positive feedback and for his/her encouraging words.

Reviewer #3 (Remarks to the Author):

In this study the authors claimed that RBD specific ACE2 binding site targeting antibodies are more sensitive to variants of concern (VOCs), whereas non-ACE2bs RBD targeting antibodies are not as sensitive to VOCs. Using a set of patient-derived monoclonal antibodies the authors demonstrate this through ELISA binding assays, cell-based flow cytometry assays, and neutralization assays. They further structurally characterize two of their nine antibodies, one of which is an ACE2bs antibody that is somewhat variant resistant, and the other, a unique antibody that targets a quaternary epitope devised of the junctions on two adjacent RBDs in the context of the Spike trimer. Altogether this study adds to the multitude of different RBD specific monoclonals that have been described and characterized, suggesting that though the RBD is a relatively small part of the Spike protein there seems to be many different modes of binding that can result in neutralization, particularly by both directly and indirectly affecting ACE-2 binding.

We thank the Reviewer for this summary and for putting our study in the context of what has been done in the field.

However, the following points should be addressed.

1. It would be informative if the authors ran neutralization tests for the non-ACE2bs mAbs TAU-1109 and -2310 against other sarbecoviruses such as SARS-CoV-1, WIV1-CoV, and SHC014-CoV to see if their breadth extends beyond just SARS-CoV-2. Since these antibodies appear to be the most variant resistant, therefore the most clinically useful, structurally they are probably the most interesting targets, and there has been no further characterization of these class of antibodies in the paper. Have there been attempts to solve the structure of at least one of these antibodies in complex with Spike or RBD?

We thank the Reviewer for bringing this up. We acquired Fab1109-spike and Fab2310-spike complex but unfortunately failed to solve the structure since the complex particles tend to disassociate under cryo-condition.

2. The authors use Pisa modelling in figure 4d to show how the substitutions found in the Beta VOC would affect the binding of TAU-2303 to the RBD. Since the changes in binding energy are quite small it's uncertain how meaningful these measurements are. It should be possible to show this how TAU-2303 accommodates these substitutions with an actual structure of the Beta RBD and TAU2303 complex. SPR or some other binding affinity measurement, might also help to pinpoint differences in binding affinity of

May 2022

TAU2303 with beta and other VOCs.

We thank the Reviewer for bringing up this key point. To address it, we performed SPR measurements of TAU-2303 with VOCs Alpha, Beta, Gamma, Delta and Omicron. This was now added to Extended Data Figure 3 panels c and d. Also, the following text was added to Line 218:

Since these proposed Van der Waals interactions would not be able to fully compensate for the lost hydrogen bonds, mutant N501Y is predicted to have a slightly lower binding affinity to Fab2303, as is confirmed by affinity measurements by Surface Plasmon Resonance (SPR) (Extended Data Figure 3c and d).

3. I am not sure how the data supports this statement: “TAU-2212 acts by binding and stabilizing the SARS-CoV-2 spike trimer in a “down” orientation and prevents conformational change to “up” RBD which is required for spike:ACE2 interactions.” It might be better to elaborate on what “small” means in terms of the rotational change towards the trimer axis and whether this is a big contributor to stabilization? The authors also don’t explore the context in which the RBD loops are disordered - is that just the case in the RBD when expressed alone, or is that true in the full-length trimer, and if so, are they ordered only upon ACE2 binding and otherwise disordered in all conformations? I think a way to investigate this mechanism would be an ELISA where you look for binding to a class 4 epitope such as CR3022 to the Spike, in the presence of binding of TAU-2122. If the Trimer is locked in the down conformation, then CR3022 can’t bind since its epitope is only exposed when the RBD is in the up conformation.

We thank the Reviewer for the question. The rotational change means the RBDs are more compact. To clarify this issue we updated the text (Line 288), and added a distance shift in Extended Data Fig. 9a

Conformation 4 of the mAb TAU-2212:spike complex has all the RBDs in the “down” position with three TAU-2212 moieties binding near the junctions between the RBDs. Each Fab2212 binds and crosslinks two adjacent “down” RBDs (RBD1 and 2) with a buried surface of 823 Å² on RBD1 and 298 Å² on RBD2 (Fig. 5b). Comparing the structures of the TAU-2212-bound and free spikes revealed that TAU-2212 binding induces conformational changes in the RBDs, producing an anti-clockwise rotation of 4.1° towards the symmetry axis, and resulting in a more compact RBD head structure after TAU-2212 binding (Extended Data Fig. 9a). The conformational changes in the RBDs bring residues at the interfaces closer and may increase the Van Der Waals interactions between the RBDs. However, no additional contacts are promoted in between the “down” RBDs. In addition, the RBD loops 454-462 and 468-489 that are disordered in free spikes are well ordered in the complex structure by forming three hydrogen bonds with the bound TAU-2212 (Extended Data Fig. 9b, Extended Data Table 5). These loops are also ordered and visible upon ACE2 binding. These data suggest that TAU-2212 acts by binding and stabilizing the SARS-CoV-2 spike trimer in a “down” orientation and prevents conformational change to “up” RBD which is required for spike:ACE2 interactions.

References:

- 1 Turner, J. S. *et al.* SARS-CoV-2 mRNA vaccines induce persistent human germinal centre responses. *Nature* **596**, 109-113, doi:10.1038/s41586-021-03738-2 (2021).
- 2 Schmitz, A. J. *et al.* A vaccine-induced public antibody protects against SARS-CoV-2 and emerging variants. *Immunity* **54**, 2159-2166 e2156, doi:10.1016/j.immuni.2021.08.013 (2021).
- 3 Zhou, Y. *et al.* Enhancement versus neutralization by SARS-CoV-2 antibodies from a convalescent donor associates with distinct epitopes on the RBD. *Cell Rep* **34**, 108699, doi:10.1016/j.celrep.2021.108699 (2021).
- 4 Mor, M. *et al.* Multi-clonal SARS-CoV-2 neutralization by antibodies isolated from severe COVID-19 convalescent donors. *PLoS Pathog* **17**, e1009165, doi:10.1371/journal.ppat.1009165 (2021).

REVIEWERS' COMMENTS:

Reviewer #1 (Remarks to the Author):

I thank the authors for providing a clear and comprehensive response to the comments on the initial submission. I am pleased to recommend this manuscript for publication in its current form.